# Transcriptomic and anatomic parcellation of 5-HT$_{3A}$R expressing cortical interneuron subtypes revealed by single-cell RNA sequencing

Sarah Frazer[1,2,*], Julien Prados[1,2,*], Mathieu Niquille[1,2], Christelle Cadilhac[1,2], Foivos Markopoulos[2], Lucia Gomez[1,2], Ugo Tomasello[1,2], Ludovic Telley[2], Anthony Holtmaat[2], Denis Jabaudon[2] & Alexandre Dayer[1,2]

Cortical GABAergic interneurons constitute a highly diverse population of inhibitory neurons that are key regulators of cortical microcircuit function. An important and heterogeneous group of cortical interneurons specifically expresses the serotonin receptor 3A (5-HT$_{3A}$R) but how this diversity emerges during development is poorly understood. Here we use single-cell transcriptomics to identify gene expression patterns operating in *Htr3a*-GFP + interneurons during early steps of cortical circuit assembly. We identify three main molecular types of *Htr3a*-GFP + interneurons, each displaying distinct developmental dynamics of gene expression. The transcription factor *Meis2* is specifically enriched in a type of *Htr3a*-GFP + interneurons largely confined to the cortical white matter. These MEIS2-expressing inter-neurons appear to originate from a restricted region located at the embryonic pallial–subpallial boundary. Overall, this study identifies MEIS2 as a subclass-specific marker for 5-HT$_{3A}$R-containing interstitial interneurons and demonstrates that the transcriptional and anatomical parcellation of cortical interneurons is developmentally coupled.

[1] Department of Psychiatry, University of Geneva Medical School, Geneva 4 CH-1211, Switzerland. [2] Department of Basic Neurosciences, University of Geneva Medical School, Geneva 4 CH-1211, Switzerland. * These authors contributed equally to this work. Correspondence and requests for materials should be addressed to A.D. (email: alexandre.dayer@unige.ch).

Cortical GABAergic interneurons (INs) constitute a highly heterogeneous population of local inhibitory neurons that control cortical microcircuit function[1–3]. Attempts to classify cortical INs are based on differences in connectivity, gene expression, physiological properties[4] and, more recently, on their functional recruitment during specific behavioural states[2]. One promising approach to parse out this complexity has been to classify cortical INs based on their developmental origins[1–6]. The majority of cortical INs is generated in the medial ganglionic eminence (MGE) of the subpallium[7–9] giving rise to parvalbumin-expressing fast-spiking basket cells, chandelier cells, and to a diverse group of somatostatin-expressing INs, which includes Martinotti cells[10–14]. The remaining fraction of cortical INs derives from the caudal ganglionic eminences (CGE)[15,16] and, to a lesser extent, from the preoptic area[17]. CGE-derived INs specifically express the serotonin receptor 3A (5-HT$_{3A}$R) in contrast to MGE-derived INs[18–20], display different types of morphologies and intrinsic electrophysiological profiles[19] and express several transcription factors such as PROX1 (refs 21–23) and COUPTFII (refs 23–25) in addition to a diversity of neurochemical markers including reelin (RELN), calretinin, neuropeptide Y, and vasointestinal peptide (VIP)[18–20]. The rich variety of cortical IN subtypes emerges during development and is determined by the restricted temporal and spatial expression of transcription factors operating during early stages of IN development[1–6,12–14,22,26–33].

Recently, single-cell RNA sequencing (RNAseq) has revealed the existence of molecularly distinct subtypes of 5-HT$_{3A}$R-containing INs residing in the cortical grey matter (GM)[34,35]. Interestingly, a subpopulation of Htr3a-GFP + INs has been shown to populate the cortical white matter (WM), and displays a variety of neurochemical, electrophysiological and morphological profiles[36]. The existence of a diversity of WM interstitial INs is documented since Ramon y Cajal in mammals including humans[37–41]. Whether Htr3a-GFP + INs in the WM constitute a molecularly distinct subclass of cortical INs as compared to Htr3a-GFP + INs residing in the GM remains to be determined. Here we used single cell RNA-seq to identify the transcriptional programs delineating WM and GM Htr3a-GFP + IN subtypes during circuit assembly. For this purpose, Htr3a-GFP + INs were isolated from the developing cortex using fluorescence-activated cell sorting (FACS) at embryonic day (E) 18, a stage characterized by migration into developing cortex, postnatal day (P) 2, when INs reach their appropriate laminar position and P5, when INs are integrating cortical microcircuits. Individual Htr3a-GFP + INs were microfluidically captured for single-cell RNA-Seq. Using this approach we identified a distinct subclass of Htr3a-GFP + INs largely confined to the cortical WM and characterized by the specific expression of the transcription factor Meis2. These neurons appeared to originate from a restricted MEIS2 + /PROX1- domain located outside of the CGE at the embryonic pallial-subpallial boundary (PSB). Finally, we found that the Nr2f2-enriched cortical IN type identified using single-cell RNAseq preferentially distributed in superficial cortical layer 1 and expressed the marker RELN. Altogether, these data provide evidence for a coupling between the transcriptional and anatomical parcellation of IN subtypes. They indicate that neural network assembly relies on circuit elements with distinct molecular identities that are spatially segregated.

## Results

### Single-cell RNA-seq identifies main types of Htr3a-GFP + INs.
Here we aimed to investigate the transcriptional heterogeneity of Htr3a-GFP + INs during the phase of cortical migration and early integration in cortical circuits. FACS was used to isolate Htr3a-GFP + INs located in the developing cortex at three distinct developmental stages: at E18 during the phase of cortical invasion, at P2 during laminar positioning and at P5 during early circuit integration (Fig. 1a). Following FACS isolation (Fig. 1b), single-cell RNA-seq from individual Htr3a-GFP + INs was performed using the Fluidigm C1 instrument (Fig. 1c)[35,42]. After quality control, a total of 82 cells at E18, 66 cells at P2 and 75 cells at P5 were used for data analysis (Supplementary Fig. 1). Htr3a-GFP + cells expressed IN -specific transcripts such as Dlx1, Dlx2, Dlx5, Dlx6, Gad1 and Gad2, thus confirming their IN identity (Supplementary Fig. 2). Analysis using t-Distributed Stochastic Neighbor Embedding (t-SNE)[43] revealed a first group of cells at E18 and three main types of cells (type 1–3) at postnatal ages P2 and P5 (Fig. 1d). Cluster stability analysis using random sampling combined with principal component analysis (PCA) and hierarchical clustering confirmed the existence of 3 main types of cells at P2 and P5 (Fig. 1e). Clustering was primarily based on type identity and secondarily on developmental age (Supplementary Fig. 3). These three main types of Htr3a-GFP + INs each contained molecularly distinct subtypes of cortical INs, (Supplementary Fig. 4), in line with recent findings revealing the existence of multiple subgroups of 5-HT$_{3A}$R + INs at later postnatal time points[34,35]. Cells belonging to main types of Htr3a-GFP + INs identified independently at P2 and P5 clustered together following t-SNE analysis, revealing gene expression changes within the Htr3a-GFP + INs types across time (Fig. 1f). We next used single cell differential expression (SCDE)[44] to identify type-specific transcripts and their expression dynamics across developmental time points, revealing distinct transcriptional states operating in Htr3a + IN types at each specific developmental stage (Fig. 1g, Supplementary Tables 1–12). The complete transcriptional dynamics for all expressed transcripts in each subgroup of Htr3a-GFP + INs can be found in Supplementary Data 1 and in an interactive format at http://dayerlab.unige.ch/htr3a_sc. Using this approach, we found that Meis2, a transcription factor also expressed in olfactory bulb INs[45,46] was the most enriched transcription factor in Htr3a-GFP + type 1 INs across postnatal development (Fig. 1g-i, Supplementary Table 13). Type 1 INs were enriched in transcription factors expressed in lateral ganglionic eminence (LGE)-derived progenitors such as Etv1 (refs 47,48), Pbx1 (ref. 49) and Sp8 (refs 50–53), which are also found in olfactory bulb INs (Fig. 1i, Supplementary Fig. 5). In contrast, type 2 and type 3 INs expressed CGE-enriched transcription factors such as Prox1 (refs 21,23), Nr2f2 (CouptfII)[23,25], Npas3 (refs 31,32) and Maf, a DLX1 and DLX2-dependent transcription factor expressed in cortical INs[33] (Fig. 1i, Supplementary Fig. 5). Taken together, our data provide a dynamic account of the transcriptional heterogeneity of main types of Htr3a + INs during early cortical circuit formation, and identify type 1, Meis2-expressing INs as genetically related to olfactory bulb INs and displaying low expression of transcription factors operating in CGE-derived cortical INs.

### MEIS2 + /Htr3a-GFP + INs are largely confined to the WM.
We determined the spatial distribution of MEIS2-expressing Htr3a-GFP + INs in the developing cortex by performing MEIS2 immunohistochemistry (IHC). We found that at P5 MEIS2 IHC labelled a large fraction (86 ± 2%, mean ± s.e.m.) of Htr3a-GFP + INs located in the WM, whereas only a very small fraction (2.2 ± 0.4%) of MEIS2 + /Htr3a-GFP + INs were observed in deep layers of the overlying cortex (Fig. 2a). Single-cell RNA-seq indicated that Meis2 + Htr3a + INs expressed low levels of Prox1 and Nr2f2 but increased levels of Etv1 and

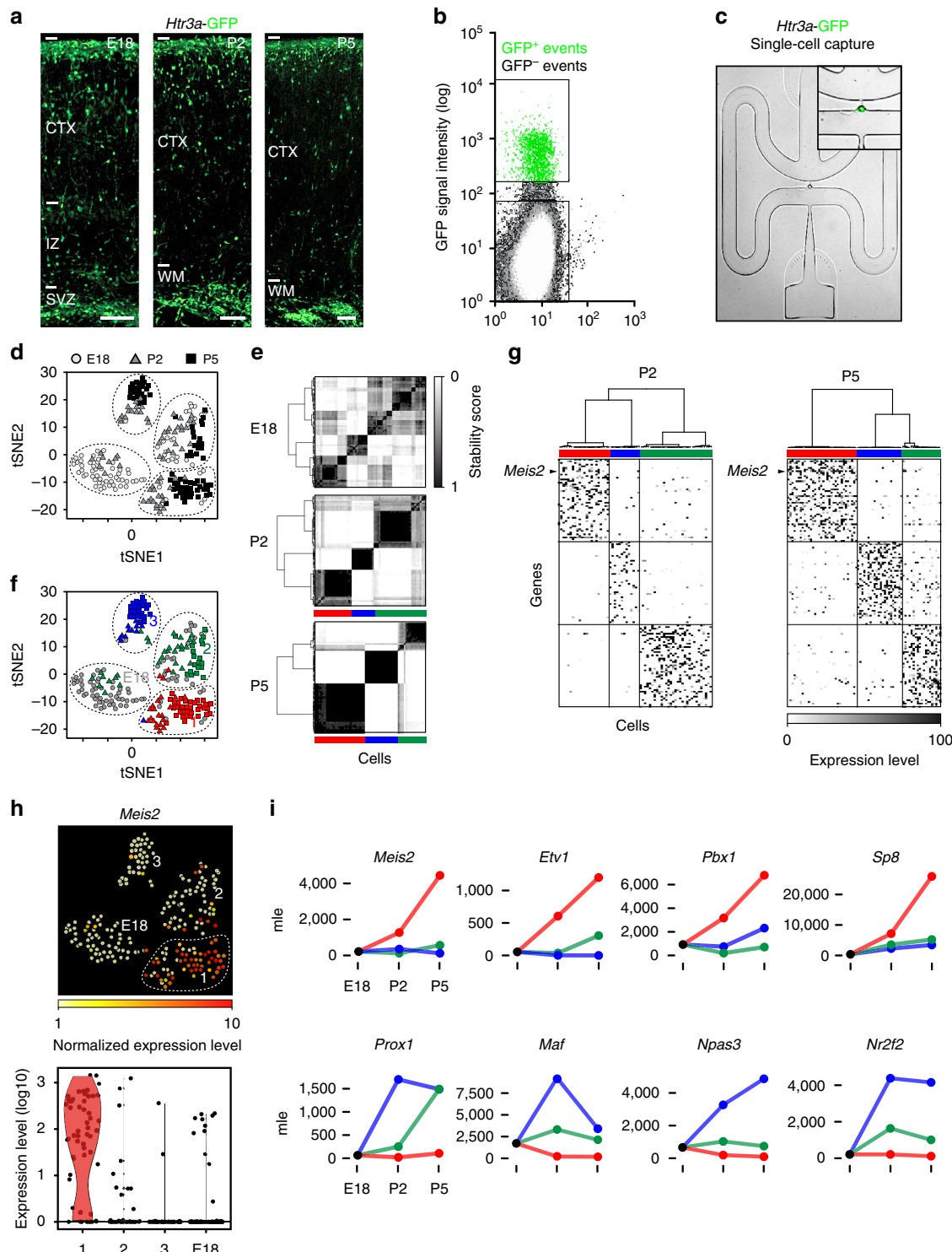

**Figure 1 | Single-cell RNA-seq identifies molecularly distinct types of *Htr3a*-GFP+ INs during development.** (**a**) Cortices and white matter from *Htr3a*-GFP+ mice were dissected at E18, P2 and P5. (**b**) *Htr3a*-GFP+ INs were isolated using fluorescence-activated cell sorting (FACS). (**c**) Individual *Htr3a*-GFP+ INs were captured in Fluidigm C1 chips and single cell RNA sequencing (RNA-seq) was performed. (*n* = 89 cells at E18 in two chips; *n* = 76 cells at P2 in one chip; *n* = 78 cells at P5 in two chips). (**d**) Density clustering by Seurat-based[43] t-Distributed Stochastic Neighbor Embedding (t-SNE) identifies three distinct groups of *Htr3a*-GFP+ INs across postnatal time-points and an E18 group. (**e**) Cluster stability analysis using random sampling combined with principal component analysis (PCA) and hierarchical clustering reveals three robust types (colour-coded) of *Htr3a*-GFP+ INs at P2 and P5. (**f**) t-SNE analysis indicates that cells belonging to *Htr3a*-GFP+ interneuron types previously identified independently at P2 and P5 (colour-coded) cluster together. Cells are colour-coded according to cluster assignment obtained using cluster stability analysis. (**g**) Heat maps displaying top 50 type-enriched genes identified using single-cell differential display (SDCE)[44] at P2 and P5. (**h**) Violin plots reveal enriched expression of *Meis2* in type 1 INs. (**i**) Time-course expression of interneuron-expressed transcription factors enriched in type 1 INs (*Meis2*, *Etv1*, *Pbx1* and *Sp8*) and types 2 and 3 (*Prox1*, *Maf*, *Npas3* and *Nr2f2*). SVZ: subventricular zone, IZ: intermediate zone, CTX: cortex, WM: white matter, mle: maximum likelihood estimates. Gene expression levels were obtained using the joint posterior estimation from the SCDE R package. Scale bars: (**a**) 150 μm at E18, 200 μm at P2 and P5.

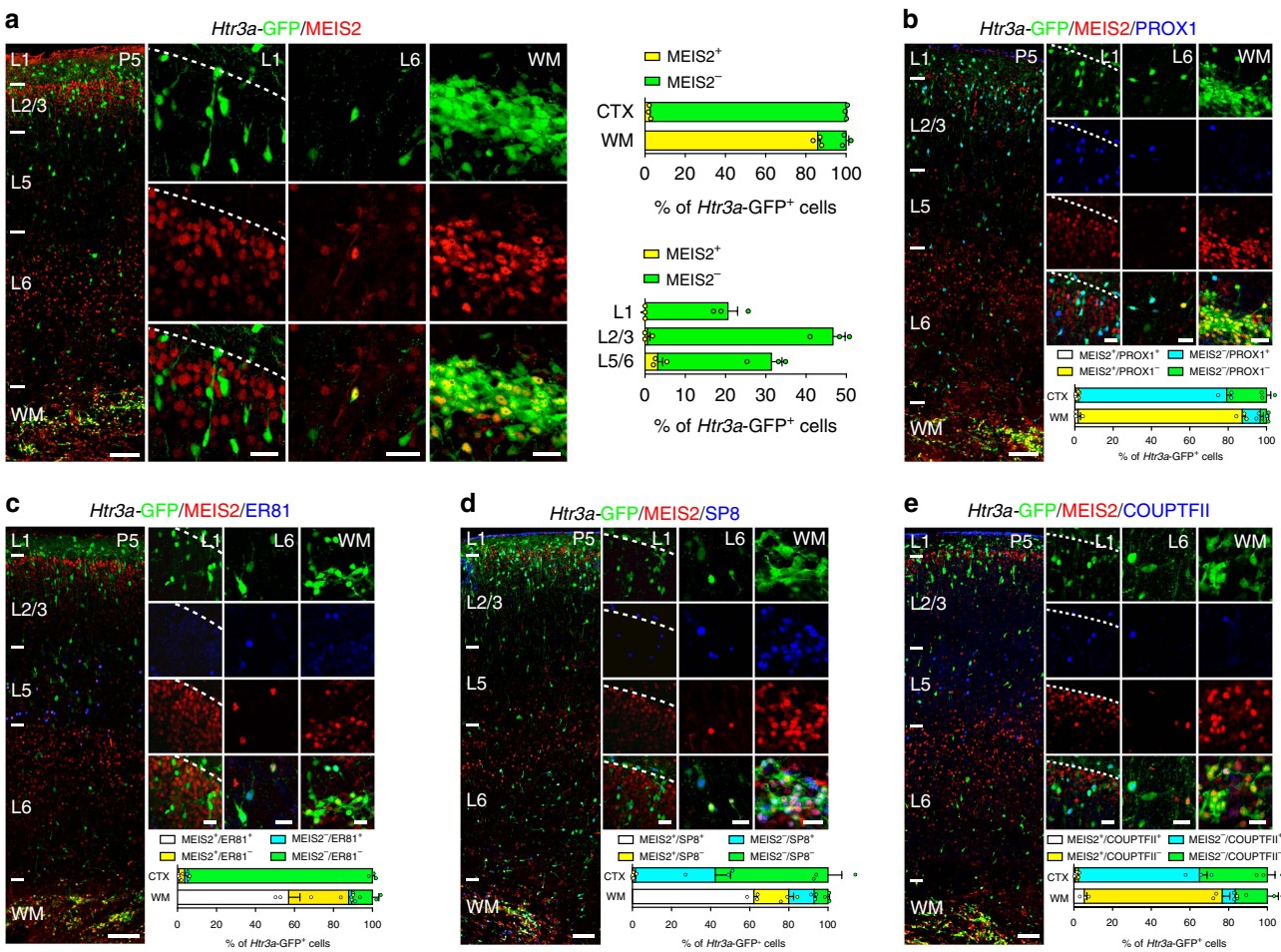

**Figure 2 | White matter (WM) *Htr3a*-GFP+ INs express MEIS2 at an early postnatal age.** (**a**) At P5, MEIS2 expression is largely confined to *Htr3a*-GFP+ INs located in the white matter (upper graph). Only a small fraction of MEIS2+/*Htr3a*-GFP+ INs are found in deep cortical layers 5/6 (lower graph). (**b**) *Htr3a*-GFP+ INs in WM express MEIS2 but not PROX1. (**c**) A large fraction of MEIS2+/*Htr3a*-GFP+ INs in the WM co-express ER81. (**d**) A large fraction of MEIS2+/*Htr3a*-GFP+ INs in the WM co-express SP8. (**e**) MEIS2+/*Htr3a*-GFP+ INs in the WM rarely co-express COUPTFII. CTX: cortex. Graphs display mean ± s.e.m. and $n = 3$ brains for each. Scale bars, (**a**–**e**) 100 μm for low-magnification images; (**a**–**e**) 20 μm for high-magnification images.

*Sp8* compared to other subgroups of *Htr3a*-GFP+ INs (Fig. 1i). We thus investigated the expression pattern of these transcription factors in MEIS2+/*Htr3a*-GFP+ INs using IHC. We found that the largest fraction of *Htr3a*-GFP+ INs located at P5 in the WM expressed MEIS2 but not PROX1 and that MEIS2 and PROX1 were almost never co-expressed (Fig. 2b). Conversely, the majority of GM cortical *Htr3a*-GFP+ INs expressed PROX1 but not MEIS2 with essentially no overlap between the two markers (Fig. 2b). IHC for ER81 indicated that a substantial fraction of *Htr3a*-GFP+ INs in the WM co-expressed MEIS2 and ER81 (Fig. 2c). Analysis of the expression of SP8 and COUPTFII revealed that a large fraction of MEIS2+/*Htr3a*-GFP+ INs in the WM expressed SP8 (Fig. 2d) but only rarely COUPTFII (Fig. 2e). In addition, MEIS2+/*Htr3a*-GFP+ INs expressing SP8 but not COUPTFII were detected in deep cortical layers (Fig. 2d,e). Taken together, these results are in accordance with the single-cell RNA-seq data and indicate that during the first postnatal week MEIS2+/*Htr3a*-GFP+ INs are mainly confined to the WM, express SP8 and ER81, but rarely PROX1 or COUPTFII.

To determine whether MEIS2+ INs remain in the WM or progressively migrate into the cortex during postnatal development, we investigated the expression of MEIS2+ in *Htr3a*-GFP+ INs at P21. As observed at P5, the majority (82 ± 1%) of *Htr3a*-GFP+ INs in the P21 WM expressed MEIS2 whereas only a very small fraction of *Htr3a*-GFP+ INs expressing MEIS2 was detected in deep cortical layers (2.1 ± 0.6%; Fig. 3a). MEIS2+/*Htr3a*-GFP+ INs in the P21 WM expressed SP8 and ER81 but not PROX1 or COUPTFII (Fig. 3b). In addition, the small fraction of MEIS2+/*Htr3a*-GFP+ INs detected in deep cortical layers co-expressed ER81, SP8 but not PROX1 or COUPTFII (Fig. 3c). Finally, whole-cell recordings of *Htr3a*-GFP+ INs located in the WM combined with IHC, indicated that these neurons displayed spontaneous inhibitory and excitatory currents (Fig. 3d–g, Supplementary Fig. 6), expressed MEIS2 and extended processes in the WM, deep layers of the cortex and subcortical structures such as the striatum (Fig. 3h, Supplementary Fig. 6). Taken together, our data identify MEIS2-expressing *Htr3a*-GFP+ INs that remain largely confined to the WM and that are functionally integrated in neuronal networks.

**MEIS2 is expressed in *Htr3a*-GFP+ INs located in the PSB.** We next investigated the developmental origin of MEIS2+/*Htr3a*-GFP+ INs. At E14, we found that

MEIS2 + /*Htr3a*-GFP + neurons were restricted to the PSB (Fig. 4a), where about one third of *Htr3a*-GFP + INs at the PSB expressed MEIS2. In contrast, in the CGE region *Htr3a*-GFP + INs only very rarely expressed MEIS (Fig. 4a). Strikingly, MEIS2 and PROX1 labelled distinct types of

*Htr3a*-GFP + INs with minimal co-expression of the two markers in the PSB and CGE (Fig. 4a). Furthermore, and as observed at postnatal time-points, MEIS2 + /*Htr3a*-GFP + INs at the E14 PSB largely co-expressed SP8 but only rarely COUPTFII (Supplementary Fig. 7a,b). At E18, a large fraction of

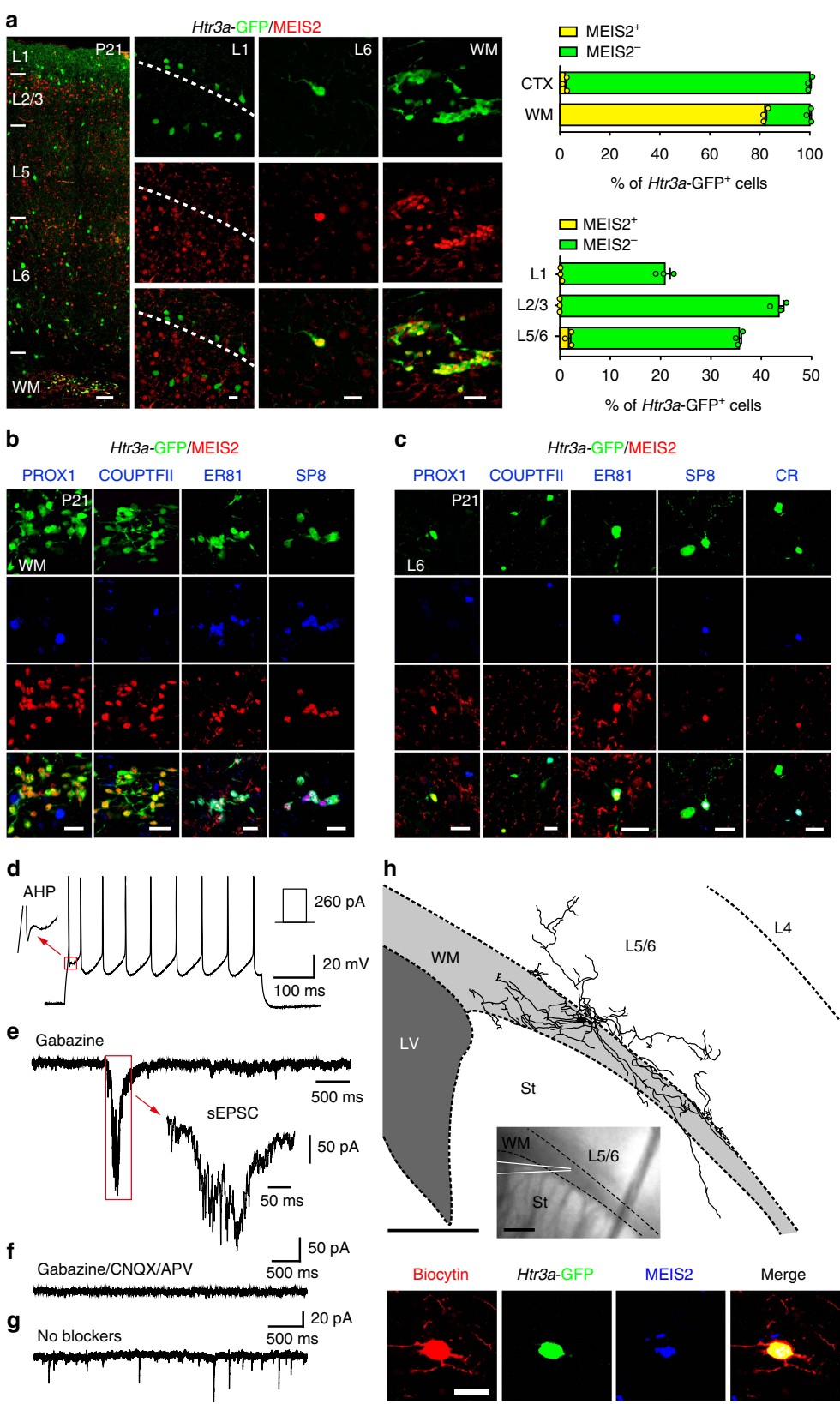

*Htr3a*-GFP + INs located at the PSB expressed MEIS2 but not PROX1 with essentially no overlap between the two populations (Fig. 4b). Importantly, a fraction of MEIS2 + / PROX1-/*Htr3a*-GFP + INs was observed in the subventricular zone (SVZ) tangential migratory stream and in the intermediate zone (IZ; Fig. 4b), suggesting that these cells are migrating into the developing pallium. MEIS2 + /*Htr3a*-GFP + INs located at the PSB, in the SVZ migratory stream and in the IZ co-expressed ER81 but more rarely COUPTFII (Supplementary Fig. 7c–e). Intraperitoneal injections of the thymidine analogue BrdU in pregnant dams between E14 and E18 indicated that *Htr3a*-GFP + INs populating the P5 postnatal WM were generated during the late embryonic period with a peak at E17 (Fig. 4c). To further investigate the origin of WM INs, isochronic grafts derived from the embryonic PSB or CGE were performed on E14 cortical slices (Fig. 4d). At day *in vitro* 2 (DIV2), PSB-derived *Htr3a*-GFP + INs migrated into the prospective cortical WM and expressed MEIS2 but not PROX1 in contrast to CGE-derived INs (Fig. 4e). In addition, confocal time-lapse imaging revealed that *Htr3a*-GFP + INs originating from the PSB displayed distinct migratory dynamics as compared to CGE-derived *Htr3a*-GFP + INs (Fig. 4f, Supplementary Movies 1, 2). Finally, *in utero* electroporation at E14 targeted to the CGE was used to ectopically express MEIS2 in CGE-derived INs. Quantification at P5 indicated that CGE-derived cortical INs, ectopically expressing MEIS2, significantly shifted their laminar distribution towards the WM and lower cortical layers as compared to controls (Supplementary Fig. 8), thus suggesting that MEIS2 regulates anatomical parcellation of cortical INs.

**COUPTFII + /*Htr3a*-GFP + INs are preferentially located in L1.** Type 1 *Meis2*-enriched *Htr3a*-GFP + INs were largely confined to the WM. In accordance with this finding, *Meis2* expression was not detected in previously identified IN subgroups located in the GM and expressing the *Htr3a* transcript[34,35] (Supplementary Fig. 9). Given the anatomical segregation of type 1 *Meis2*-enriched INs, we next investigated whether other main types of INs displayed evidence for anatomical parcellation. Single-cell RNA-seq indicated that type 3 *Htr3a*-GFP + INs were enriched in the transcription factor *Nr2f2* (Fig. 1i). Using IHC to analyse the spatial distribution of *Htr3a*-GFP + INs expressing COUPTFII in the cortex, we found that COUPTFII + / *Htr3a*-GFP + INs were preferentially located in layer 1 (L1) at P5 and P21 (Fig. 5a,b). *Htr3a*-GFP + INs in the GM have been divided in largely mutually exclusive subtypes based on the expression of RELN or VIP[3,16,18,19,34,35]. Interrogation of our single-cell data indicated that type 3 *Nr2f2*-enriched *Htr3a*-GFP + expressed higher levels of the *Reln* transcript as compared to other types (Fig. 5c). In accordance with this observation, IHC indicated that RELN was preferentially expressed in COUP-TFII + /*Htr3a*-GFP + INs as compared to COUPTFII-/*Htr3a*-GFP + INs (Fig. 5d). In addition, cross-comparison with previously published single-cell RNAseq data

sets[34,35] indicated that type 3 *Htr3a*-GFP + INs were preferentially enriched in transcripts found in *Reln* expressing subtypes as compared to *Vip* expressing subtypes (Supplementary Fig. 10). Overall, these results further support a coupling between anatomical and transcriptional parcellation of *Htr3a*-GFP + IN types.

## Discussion

*Htr3a*-GFP + INs mainly derive from CGE- but not MGE-derived progenitors[3,18–20]. These cells constitute a heterogeneous group of INs that display a variety of neuro-chemical, morphological and electrophysiological features[17–25]. Here we aimed to determine whether transcriptional states could identify cardinal types of *Htr3a*-GFP + INs during early cortical circuit formation. To access these programs, *Htr3a*-GFP + INs were isolated at E18, P2 and P5 and single-cell RNA-sequencing was performed using the C1 Fluidigm system. Using this strategy, we found that 3 molecularly distinct types of *Htr3a*-GFP + cortical INs emerge during the first postnatal week. We next identified transcripts enriched in each given type of *Htr3a*-GFP + INs and provide transcriptomic profiles describing the developmental gene expression dynamics operating in these three main types of *Htr3a*-GFP + INs as they integrate into cortical circuits. Type 1 *Htr3a*-GFP + INs were enriched in transcription factors expressed in LGE-derived progenitors and olfactory bulb INs, such as *Meis2* (refs 45,46), *Etv1* (refs 47,48), *Pbx1* (ref. 49) and *Sp8* (refs 50–53), and were found to be largely confined to the WM. In contrast, type 2 and 3 IN types expressing low levels of *Meis2*, were enriched for transcription factors operating in CGE-derived cortical INs such as *Prox1* (refs 21,22) and *Npas3* (refs 31,32) and populated the GM. Furthermore, type 3 *Htr3a*-GFP + INs enriched in *Nr2f2* were found to be preferentially distributed in superficial cortical layer 1 and belonged to the RELN + subtype. It should be noted that these 3 main types of *Htr3a*-GFP + INs could be further subdivided in additional distinct molecular subtypes, in line with recent findings describing a variety of different subtypes of *Htr3a*-GFP + INs in the more mature cortex and white matter at transcriptomic[34,35], morphological and electrophysiological levels[19,36]. Larger data sets containing additional developmental time-points will be required to determine how the different subtypes of *Htr3a*-GFP + INs detected in the early postnatal period generate the large diversity of *Htr3a*-GFP + INs described in adulthood. Among potential mechanisms, it has been proposed that, in addition to cell-intrinsic factors, early activity could be required to further specify classes of INs into more refined subtypes of INs, thus allowing them to be finely tuned to network activity and acquire unique functional properties[2].

In the cortex, CGE-derived *Htr3a*-GFP + INs preferentially populate superficial cortical layers whereas MGE-derived INs are more frequently distributed in deeper cortical layers[3,15]. These contrasting gradients in the laminar positioning of CGE- versus MGE-derived INs suggest that distinct transcriptional programs

**Figure 3 | MEIS2 + /*Htr3a*-GFP + INs are functionally integrated in the juvenile cortical white matter (WM).** (**a**) At P21 MEIS2 expression is maintained in a large fraction of *Htr3a*-GFP + INs located in WM (upper graph; n = 3 brains) while only a small fraction of MEIS2 + /*Htr3a*-GFP + INs are found in deep cortical layers 5/6 (lower graph; n = 3 brains). (**b**) At P21, MEIS2-expressing *Htr3a*-GFP + INs in WM do not express PROX1 and COUPTFII but express ER81 and SP8 (**c**) At P21, MEIS2 + /*Htr3a*-GFP + INs located in deep cortical layers do not express PROX1 and COUPTFII but express ER81, SP8 and calretinin (CR). (**d**) Example trace of current clamp recording from a MEIS2 + /*Htr3a*-GFP + cell, showing its response to a depolarizing current injection. The magnified area depicts the triphasic after-hyperpolarization potential (AHP) observed in some (7 out of 13) recorded cells. (**e**) Example trace of voltage clamp recording from the same cell, showing long-lasting excitation due to sEPSC in the presence of Gabazine. (**f**) The sEPSCs were abolished in the presence of CNQX and APV respectively. (**g**) Example trace of voltage clamp recording from a different MEIS2 + /*Htr3a*-GFP + cell, showing mixed sEPSCs and sIPSCs in the absence of blockers. (**h**) Reconstruction of WM *Htr3a*-GFP + recorded cell in (**e**) expressing MEIS2 and extending processes in the white matter, overlying cortical layers 5/6 and striatum (St). LV: lateral ventricle. Graphs display mean ± s.e.m. Scale bars, (**a**) 100 µm for low-magnification image; (**a–c,h**) 20 µm for high-magnification images; (**h**) 200 µm for reconstruction and low-magnification insert image.

operating during early circuit assembly instruct the anatomical parcellation of IN subtypes[1–3,5,6]. Here we find that type 1 *Meis2*-enriched *Htr3a*-GFP + INs identified using single-cell RNAseq are largely confined to the cortical WM, whereas type 3 *Nr2f2*-enriched *Htr3a*-GFP + INs are preferentially distributed in superficial layer 1. Overall, these results provide evidence that

the transcriptomic identity of an IN type is coupled to its final anatomical location.

Neurons residing in the cortical WM have been observed in several different types of mammals including humans[36–41]. Neurochemical, morphological and electrophysiological analysis revealed that WM interstitial INs are highly diverse[36–41]. During

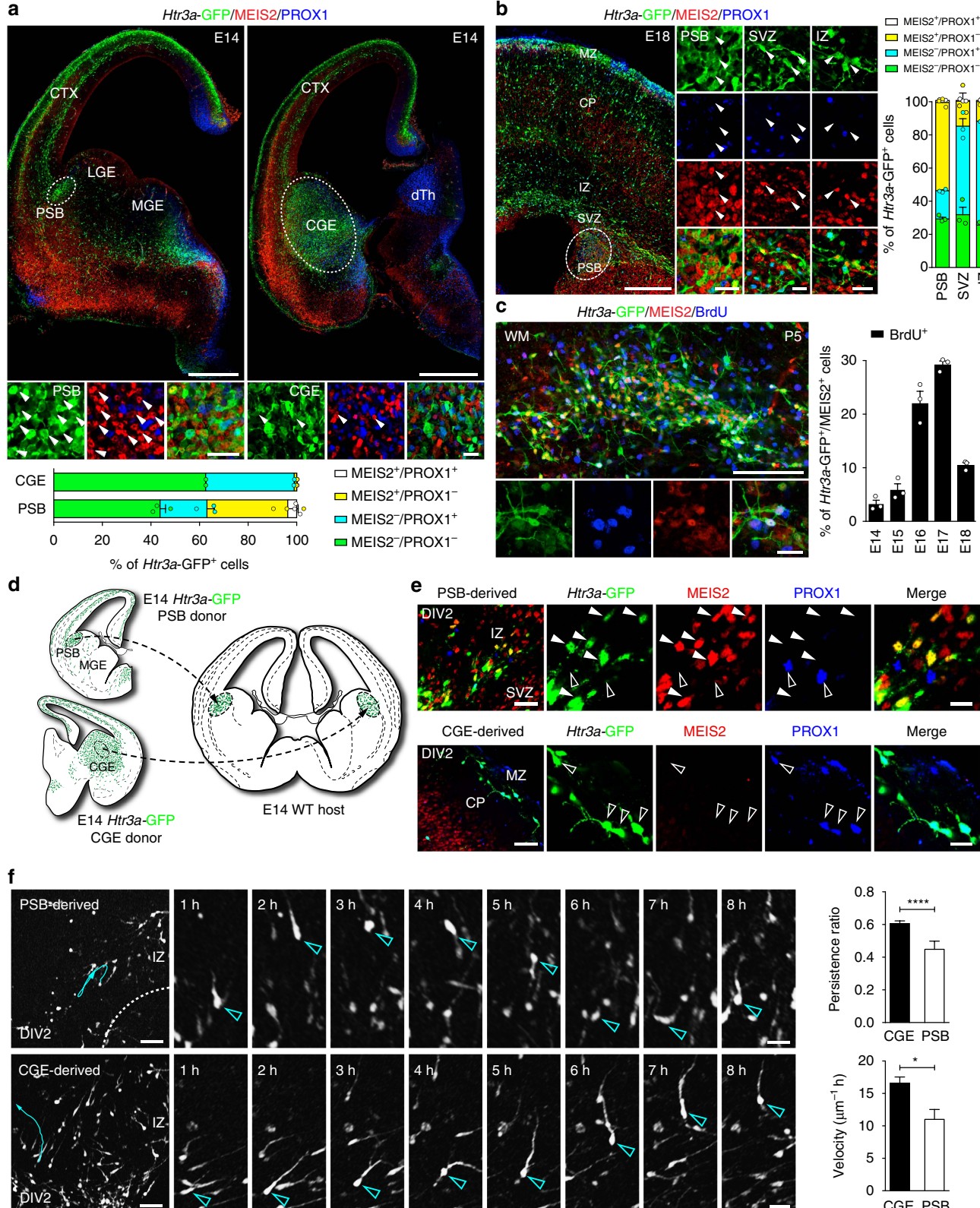

the first postnatal week, *Htr3a*-GFP+ INs located in the cortical WM constitute a relatively large pool of INs in contrast to MGE-derived INs[54]. In adulthood, WM interstitial *Htr3a*-GFP+ INs have been reported to extend processes into the overlying deep cortical layers and subcortical structures such as the hippocampus and striatum[36]. In addition, patch-clamp recordings indicated that *Htr3a*-GFP+ INs received excitatory and inhibitory inputs from cortical and subcortical structures and inhibited principal cells of the adjacent cortex[36]. Future studies using optogenetic-based approaches are thus necessary to probe the role of this WM IN type in cortical circuit function. Here we characterized the gene expression patterns expressed in WM *Htr3a*-GFP+ INs and revealed that they specifically express the transcription factor MEIS2 in contrast to *Htr3a*-GFP+ INs located in the cortical GM. Strikingly, WM *Htr3a*-GFP+ INs rarely expressed CGE-enriched transcription factors such as PROX1 (refs 20,26) or COUPTFII (refs 29,32), but expressed transcription factors enriched in olfactory bulb INs such as ER81 (refs 39–41) and SP8 (refs 43–45). In addition, electrophysiological recordings of WM INs indicated that MEIS2+/*Htr3a*-GFP+ INs received inhibitory and excitatory inputs and extended processes in WM, overlying cortex and subcortical structures. Overall, these data indicate that cortical INs largely confined to the cortical WM express MEIS2 and not classical CGE-enriched transcription factors, and are functionally integrated into neuronal networks.

The fact that WM *Htr3a*-GFP+ INs do not express CGE-derived transcription factors suggests that this IN subtype may originate from an embryonic region located outside of the CGE. Indeed, we found that *Htr3a*-GFP+ INs belonging to the MEIS2 lineage emerge outside of the CGE from a restricted LGE region located at the border between the pallium and the subpallium. It is generally assumed that LGE-derived progenitors exiting the subpallium migrate in the rostral migratory stream to populate the olfactory bulbs but do not contribute to cortical IN subtypes[1–8]. However, initial studies using cortical slices and explants have suggested that a fraction of LGE progenitors may reach the pallium[9,55]. Here we found that MEIS2+/*Htr3a*-GFP+ INs exit the embryonic PSB region, migrate in the SVZ tangential migratory stream and IZ and display distinct migratory dynamics as compared to CGE-derived *Htr3a*-GFP+ INs. Strikingly, in the PSB region and in the tangential migratory stream, MEIS2+/*Htr3a*-GFP+ INs rarely co-expressed the CGE-enriched transcription factors PROX1 and COUPTFII but expressed ER81 and SP8, further suggesting that this type of *Htr3a*-GFP+ INs displays a distinct molecular identity as compared to CGE-derived *Htr3a*-GFP+ INs.

In conclusion, our study shows that single-cell transcriptomics allows the identification of main types of *Htr3a*-GFP+ INs as they assemble into cortical circuits. It provides a first roadmap to the developmental molecular dynamics operating in main types of *Htr3a*-GFP+ INs and identifies MEIS2 as a marker for *Htr3a*-GFP+ INs largely confined to the cortical WM. It demonstrates the existence of a strong coupling between the transcriptomic and anatomical parcellation of cortical IN subtypes. Overall, it provides further evidence that neural network assembly requires circuit elements with distinct molecular identities that are spatially segregated.

## Methods

**Mice.** Animal experiments were performed according to international and Swiss guidelines and approved by the Geneva local animal care committee. Timed-pregnant mice were obtained by overnight mating. To label INs preferentially derived from the CGE, we used transgenic mice of both sexes expressing the enhanced green fluorescent protein (eGFP) under the control of the *Htr3a* regulatory sequences (*Htr3a*-GFP)[20]. Mice were maintained on a C57BL/6 background.

**FACS and single-cell capture.** Brains were extracted in ice cold Hanks' balanced Salt Solution (HBSS, Sigma) from *Htr3a*-GFP+ embryos at E18, from pups at P2 and P5. Each time point consisted of pooled embryos or pups (n = 4–5). Cortices were isolated under a fluorescence dissecting scope (Leica, M165 FC). E18 and P2 cortices were dissected in HBSS, dissociated in Pronase (Sigma) at 37 C° in Neurobasal medium (Invitrogen) for 10 min and blocked in fetal calf serum (FCS). P5 cortices were dissected in artificial cerebrospinal fluid (ACSF) containing (in mM): NaCl (120), KCl (3.5), CaCl$_2$ (2.5), MgSO$_4$ (1.3), NaH$_2$PO$_4$ (1.25), NaHCO$_3$ (25), glucose (25), continuously bubbled with 95% O2 and 5% CO2, pH 7.4, chemically dissociated in Pronase at room temperature for 30 min, and blocked with 5% FCS in ACSF. At all ages, dissociated cells were then titrated progressively with Pasteur pipettes with tips fire-polished to diameters of approximately 600, 300 and 150 μm and distribution was checked using a Tali image cytometer (ThermoFisher Scientific). *Htr3a*-GFP+ cells were isolated using a Moflo Astrios FACS (Beckman Coulter). Single cells were sorted according to their forward and side scattering properties and their level of GFP fluorescence emission. Four microliters of C1 Suspension Reagent (Fluidigm) was added to 16 μl of FACS isolated *Htr3a*-GFP+ cells (300–500 cells per μl cell suspension). The 20 μl of the cell suspension mix was loaded on a C1 Single-Cell AutoPrep IFC microfluidic chip designed for 5–10 μm cells, and the chip was then processed (C1 system, Fluidigm) following the manufacturer's protocol. The plate was then transferred to an inverted microscope to validate the position of single *Htr3a*-GFP+ cells within each capture site. Lysis, reverse transcription and PCR were then performed directly on the chip using SMARTer Ultra Low RNA Kit for the C1 System (#634833,Takara Clontech). On termination of the run, amplified cDNA was collected and the concentration of cDNA was assessed on a SpectraMax Gemini Fluorimeter (Molecular Device). The typical yield was about 0.8 ng ml$^{-1}$ per cell.

**RNA-seq library preparation and sequencing.** RNA sequencing was performed by the Genomics Core Facility of the University of Geneva. RNA libraries were prepared using Illumina Nextera XT DNA Sample Preparation kit. Libraries were multiplexed and sequenced on an Illumina HiSEQ2000 with a setup that generates pair-end reads of 2 × 50 bases pairs (bp) at an expected depth of 10 million read-pairs per library. Reads were aligned on the mouse genome (assembly GRCm38, annotation from Ensembl v78) using Tophat v2.0.12 (ref. 56), and transcripts were quantified with HTSeq v0.6.1 (ref. 57). Read counts were

**Figure 4 | MEIS2+/*Htr3a*-GFP+ INs are found at the pallial-subpallial boundary (PSB) and in the tangential migratory stream during embryonic development.** (**a**) At E14, MEIS2 is expressed in PROX1-/*Htr3a*-GFP+ INs located at the PSB at the level of lateral ganglionic eminence (LGE) (arrowheads). At the level of the caudal ganglionic eminence (CGE), *Htr3a*-GFP+ INs express PROX1 but only very rarely MEIS2 (n = 3 brains). (**b**) At E18, MEIS2 is expressed in PROX1-/*Htr3a*-GFP+ INs located at the PSB, in the subventricular zone (SVZ) tangential migratory stream and in the intermediate zone (IZ) (arrowheads; n = 3 brains). (**c**) MEIS2+/*Htr3a*-GFP+ INs populating the P5 white matter (WM) were co-labelled with BrdU injected between E14 and E18 (n = 3 brains for each injection time point). (**d**) Isochronic grafts of *Htr3a*-GFP+ INs on E14 cortical slices were performed using PSB or CGE tissues isolated by microdissection. At day *in vitro* 2 (DIV2), PSB-derived and CGE-derived *Htr3a*-GFP+ INs were assessed in the developing cortex using IHC and time-lapse imaging. (**e**) PSB-derived *Htr3a*-GFP+ INs located in the IZ are MEIS2+ (arrowheads) but not PROX1+ (open arrowheads). In contrast, CGE-derived *Htr3a*-GFP+ INs located in MZ and CP are PROX1+ (open arrowheads) but not MEIS2+. (**f**) Time-lapse imaging reveals that PSB-derived *Htr3a*-GFP+ INs display a significantly lower migratory persistence ratio and speed as compared to CGE-derived *Htr3a*-GFP+ INs (****$P < 0.001$, *$P < 0.05$, unpaired Student's *t*-test, n = 47 PSB-derived cells in three slices from three brains and n = 156 CGE-derived cells in three slices from three brains). Migratory path of cells (cyan arrows) displayed in the time-lapse sequence are depicted in blue in the low magnification image. Graphs display mean ± s.e.m. CTX: cortex, dTh: dorsal thalamus, MZ: marginal zone, CP: cortical plate. Scale bars, (**a**) 500 μm for low-magnification image; (**b**) 250 μm for low-magnification image; (**c**) 100 μm for low-magnification image; (**f**) 50 μm for low-magnification images; (**a**–**f**) 20 μm for high-magnification images.

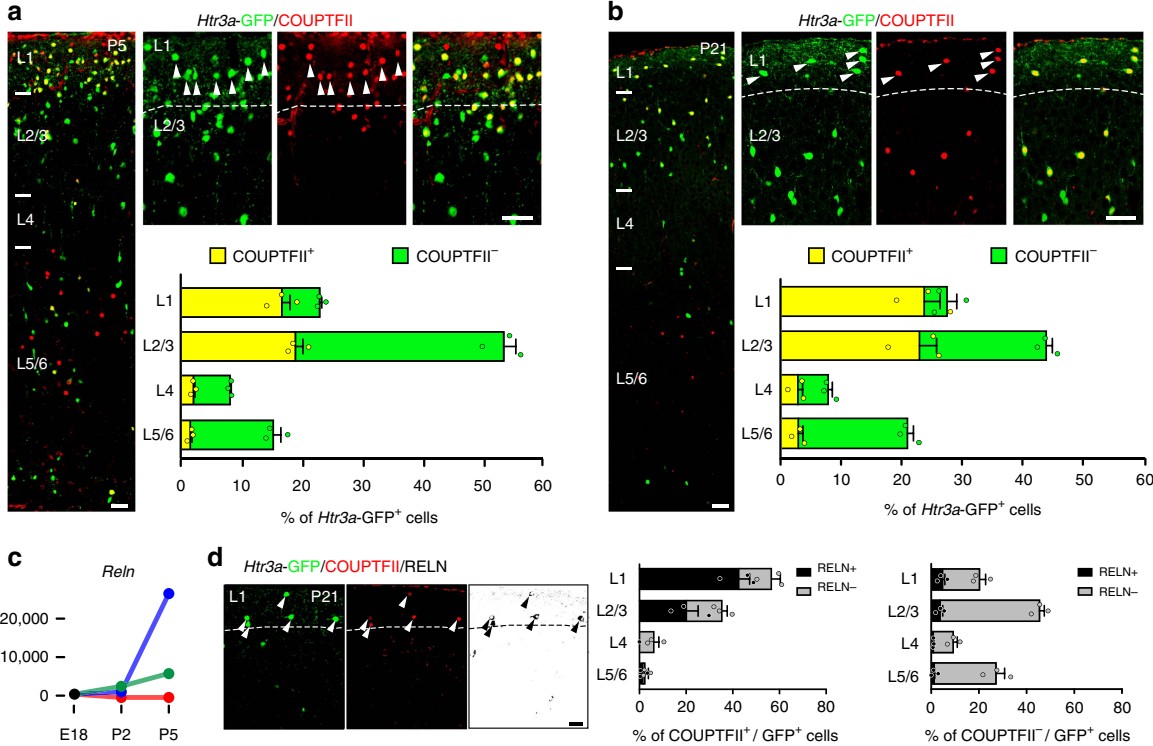

**Figure 5 | *Htr3a*-GFP+ INs expressing COUPTFII are preferentially distributed in cortical layer 1 and express reelin (RELN). (a)** At P5, COUPTFII is preferentially expressed in *Htr3a*-GFP+ INs located in cortical layer 1 (L1) (arrowheads). **(b)** Preferential expression of COUPTFII in *Htr3a*-GFP+ INs located in cortical L1 is maintained at P21 (arrowheads). **(c)** The *Reln* transcript is enriched in *Nr2f2*-expressing type 3 (blue) *Htr3a*-GFP+ INs **(d)** RELN is preferentially expressed in COUPTFII+/*Htr3a*-GFP+ INs located in L1 (arrowheads) compared to COUPTFII-/*Htr3a*-GFP+ INs. Graphs display mean ± s.e.m. and $n = 3$ brains for each. Scale bars, **(a,b)** 50 μm for low-magnification images, **(a,b,d)** 25 μm for high-magnification images.

further normalized into reads per million (r.p.m.), dividing the counts by the number of mapped reads, and multiplied by $10^6$. All the computations were done on Vital-It cluster administrated by the Swiss Institute of Bioinformatics.

**Quality control.** A total of 243 single cells were obtained using microfluidic chips at the three developmental ages ($n = 89$ at E18, two chips; $n = 76$ at P2, one chip; $n = 78$ at P5, two chips). Quality control was assessed using the following criteria: at least 5 million read-pairs were sequenced per cell; more than 80% of the read-pairs were mapped to the genome; less than 5% of the reads were mapped to the mitochondrial chromosome; at least one read was aligned to eGFP. A total of 223 cells (82 E18 cells, 66 P2 cells, 75 P5 cells) satisfied all above quality criteria. All cells had at least ten sequence reads aligned to at least one of the following IN markers: *Dlx1, Dlx2, Dlx5, Dlx6, Gad1* and *Gad2*.

**Identification of *Htr3a*-GFP+ IN types.** To identify clusters of cells, we performed PCA and cluster analysis independently at E18, P2 and P5. As input, we used RNA expression of genes expressed above 10 RPM in more than 5% of the cells. Principal component analysis (PCA) followed by a hierarchical clustering was used to identify clusters of cells. In both cases, we utilized the implementation available in the standard packages of the R programming language. To assess robustness of clusters, we ran the PCA and clustering procedures 100 times using 80% of the cells randomly sampled from the initial pool and calculated a stability score, ie, the proportion of time each pair of cells clustered together. A heatmap was computed based on this information and the hierarchical organization of cells in the heatmap provides the final robust assignment of cells in clusters. To delineate clusters of cells across developmental time, we used the t-SNE dimensionality reduction method available in the Seurat R package[43].

**Identification of type-enriched genes.** Single-Cell Differential Expression (SCDE)[44] was used to compute probability distribution of genes expression and determine differentially expressed genes between subgroups at each developmental time-point. SCDE provides a Maximum Likelihood Estimate (MLE) of the fold change in gene expression as well as a $Z$-score reflecting the confidence level of the over-expression or down-expression of a given gene (as a more classical $P$ value would do). Transcripts up or downregulated in a given IN type at P2 and P5 timepoints were identified based on the following criteria: a minimum fold

change in gene expression compared to other subgroups (MLE ≥ 2, or MLE ≤ − 2), and a minimum confidence level ($Z$-score ≥ 2, or $Z$-score ≤ − 2). Violin plots were obtained using SCDE-adjusted expressions of identified genes. The representative gene expression for a distinct type of INs across time-points was obtained using the MLE of the joint posterior distributions computed by SCDE.

To further investigate the molecular diversity of the three main types of *Htr3a*-GFP+ INs described in Fig. 1, additional tSNE and cluster analysis were performed on subset of cells composing the 3 main types of INs. More specifically, we used spectral tSNE with a perplexity parameter of 4 and a hierarchical clustering algorithm computed from the Euclidian distances in the tSNE space. The resulting hierarchy was cut manually according to its structure to identify subtypes of INs. Three indicative marker genes were then selected for each subtypes of INs from top scored genes resulting from SCDE (Supplementary Fig. 4).

To investigate whether type 2 or type 3 *Htr3a*-GFP+ INs shared increased percentage of enriched genes with *Reln*-enriched or *Vip*-enriched IN subtypes identified in the gray matter of the somatosensory cortex by Zeisel *et al.*[35] or Tasic *et al.*[34], we identified these different subtypes in the available databases. More specifically in the Zeisel *et al.*[35] data set, we extracted 48 cortical INs classified as belonging to *Vip*-expressing IN subtypes (Int6, Int9, Int10) and 80 cells belonging to *Reln*-expressing subtypes (Int11, Int12, Int14, Int15, Int16). Similarly, in the Tasic *et al.*[34] data set we extracted 48 cortical INs classified as belonging to *Vip*-expressing subtypes (*Vip Chat, Vip Gpc3, Vip Mybpc1, Vip Parm1, Vip Sncg*) and 33 cells belonging to *Reln*-expressing subtypes (*Ndnf Car4, Ndnf Cxcl14*). Genes differentially enriched in the *Reln* and *Vip*-expressing subtypes were obtained using SCDE. Incremental $Z$-score confidence levels by steps of 0.1 were used to determine the percentage of shared enriched genes in type 2 or type 3 INs identified in this study with *Reln* or *Vip*-expressing subtypes identified in existing databases (Supplementary Fig. 10).

The raw data resulting from single cell RNA-seq analysis are provided in Supplementary Data 1, which contains for each cell the number of read-pairs aligned to the exonic region of expressed genes, as computed by HTSeq v0.6.1, the coefficient used to normalize the data into RPM, the percentage of reads mapping to mitochondrial chromosome and to eGFP, and the results of the SCDE analysis.

**Tissue processing and immunohistochemistry.** Pregnant females were killed by lethal intraperitoneal (i.p.) injection of pentobarbital (50 mg kg $^{-1}$), embryos were collected and brains were dissected and fixed overnight (O.N.) in cold

4% paraformaldehyde (PFA) dissolved in 0.1M phosphate buffer pH 7.4. For postnatal brain collection, animals were deeply anaesthetized by i.p. injection of pentobarbital and transcardially perfused with 0.9% saline/liquemin followed by cold 4% PFA (P5). Brains were then postfixed in 4% PFA O.N. For bromodeoxyuridine (BrdU) experiments, a single i.p. injection of BrdU $(50 \, mg \, kg^{-1})$ was done in pregnant E14 to E18 dams and brains from P5 pups were analysed. 60 μm-thick coronal brain sections were cut on a Vibratome (Leica VT1000S) and slices were stored in cryoprotective solution at $-20\,°C$ or directly processed for IHC as described[20]. For BrdU IHC, sections were denatured in 2 N HCL for 60 min at $37\,°C$ (ref. 54). The following primary antibodies were used: chicken anti-GFP (1:2,000; Abcam; ab13970), rabbit anti-chicken ovalbumin upstream promoter transcription factor 2 (COUPTFII; 1:500; Abcam, ab42672), mouse anti-MEIS2 (1:2000; Sigma, WH0004212M1), goat anti-SP8 (1:50; Santa Cruz; sc-104661), goat anti-PROX1 (1:500, R&D Systems, AF2727), rabbit anti-calretinin (1:2,000; Swant, CR7697), and rabbit anti-ER81 (ref. 58) (1:5000; gift from Silvia Arber), mouse anti-RELN (1:1,000; Abcam, ab78540), rat anti-BrdU (1:100; Abcam, ab6326). Secondary goat or donkey Alexa-404 (Abcam), -488, -568 and -647 antibodies (Molecular Probes, Invitrogen) raised against the appropriate species were used at a dilution of 1:500–1,000. Alexa 594-conjugated Streptavidin (1:500; Invitrogen, S32356) was used to label recorded cells in the WM. Apart sections stained with Alexa-405, tissue was counterstained with Hoechst 33258 (1:10,000).

**Quantification of IN identity and distribution.** Images were acquired using a confocal microscope (Nikon A1R) equipped with oil-immersion 40x and 60x objectives (CFI Plan Fluor 40x/1.3 and CFI Plan Apo VC H 60x/1.4, Nikon). At E14, $Htr3a$-GFP $+$ cells were counted in the PSB region located at the level of the LGE and in the CGE. In the PSB, the co-expression of SP8/COUPTFII was analysed on MEIS2 $+$/$Htr3a$-GFP $+$ cells. At E18, co-expression of PROX1, ER81, SP8 and COUP-TFII in the PSB, in the subventricular zone (SVZ) tangential migratory stream and in the intermediate zone (IZ). Co-localization with MEIS2/PROX1 was analysed on all $Htr3a$-GFP $+$ cells whereas co-expression of ER81, SP8 and COUPTFII was analysed on MEIS2 $+$/$Htr3a$-GFP $+$ cells. At P5 and P21, $Htr3a$-GFP $+$ cells were counted in the primary motor and somatosensory cortices and in the underlying white matter (WM) in coronal sections corresponding to the levels $+0.2$ to $-0.1$ mm from Bregma. The quantified region did not contain the rostral migratory stream or the SVZ. At P5 and P21, co-expression of PROX1, ER81, SP8 and COUPTFII was analysed on $Htr3a$-GFP $+$/MEIS2 $+$ cells in the WM and cortex. At P5, BrdU $+$/ MEIS2 $+$/$Htr3a$-GFP $+$ cells were counted in WM following BrdU injections ranging from E14 to E18. Detailed counts of cells and brains for each experimental condition are provided in Supplementary Table 14.

**Electrophysiological recordings and morphology.** 200 μm-thick coronal brain slices were prepared from 3 to 4 week-old $Htr3a$-GFP mice with a vibratome (Leica VT 1000S). In the recording chamber, slices were continuously superfused with ACSF ($32\,°C$) containing (in mM): NaCl (119), KCl (2.5), CaCl$_2$ (2.5), MgSO$_4$ (1.3), 1.0 NaH$_2$PO$_4$ (1.0), NaHCO$_3$ (26.2), and glucose (22), and equilibrated with 95% O$_2$/5% CO$_2$, pH 7.4. Whole-cell recordings were obtained from visually identified $Htr3a$-GFP $+$ cells using an upright microscope (Zeiss Axioskop FS) equipped with differential interference contrast and standard epifluorescence. Borosilicate glass patch pipettes had a resistance of 5–6 MΩ when filled with an internal solution containing (in mM): K gluconate (105), KCl (30), HEPES (10), Phosphocreatine (10), Mg-ATP (4), Na-GTP (0.3), and Biocytin (8.1). Voltage clamp recordings were performed in gap-free mode by holding cells at $-70$ mV. Spontaneous excitatory postsynaptic currents (sEPSCs) were recorded in the presence of 10 μM 2-(3-Carboxypropyl)-3-amino-6-(4 methoxyphenyl)pyr-idazinium bromide (SR95531, Sigma) whereas spontaneous inhibitory postsynaptic currents (sIPSCs) were recorded in the presence of 10 μM 6-Cyano-7-nitroqui-noxaline-2,3-dione (CNQX, Sigma) and 50 μM D( − )-2-Amino-5-phosphono-pentanoic acid (D-APV, Sigma). Current clamp recordings were performed at rest and firing properties were studied by delivering consecutive current pulses, 500ms duration each, ranging from $-20$ to $+170$ pA with a 10 pA increment. Data were acquired using a Multiclamp 700B Amplifier (Molecular Devices), and digitized at 10 kHz (National Instruments), using MATLAB (MathWorks)-based Ephus software (Ephus; The Janelia Farm Research Center). Offline analysis was performed using Clampfit (Version 10.1.0.10, Molecular Devises). A total of 13 $Htr3a$-GFP $+$ cells from 7 brains were recorded in the WM and all of them exhibited spontaneous postsynaptic currents that were eventually blocked with SR95531 or CNQX and APV. Following patch-clamp recordings, slices were incubated in ACSF for 1–2 h and then fixed overnight with 4% PFA/2% Glutaraldehyde in 0.1 M phosphate buffer and directly processed for immunohis-tochemical staining for GFP, MEIS2 and Biocytin. Briefly, slices were blocked, incubated (2 days O.N., $4\,°C$) with primary antibodies in blocking solution and secondary antibodies were incubated (O.N., $4\,°C$) together with Streptavidin Alexa 594-conjugated to recover morphology. Location of the recorded cell in the WM and expression of MEIS2 was assessed using confocal microscopy. Morphological reconstructions of Biocytin-filled cells were performed with Neurolucida software (v. 11.02.1, MBF Bioscience, Microbrightfield), linked to a microscope (Nikon eclipse 80i) equipped with an oil-immersion 100 × objective

(N.A. 1.4, Nikon). Traces were extracted with Neurolucida Explorer (v. 11.02.1, MBF Bioscience, Microbrightfield). All 13 $Htr3a$-GFP $+$ recorded cells in the WM were confirmed as MEIS2 $+$.

**Grafts and time-lapse imaging.** Acute slices were prepared as described[20] from $Htr3a$-GFP $+$ (donor) and wildtype (WT) (host) brains at E14. Briefly, brains were extracted, embedded in HBSS with 3% ultra-pure low-melting point agarose (Roth) and 250 μm-thick slices were cut on a Vibratome (VT1000S; Leica) in cold HBSS. WT host slices were deposited on porous nitrocellulose (Millicell-CM, Millipore) inserts and transferred to Neurobasal medium (NBM) in an incubator ($37\,°C$ and 5% CO$_2$) for recovery at least 1 h. Microdissection of the PSB and CGE regions containing $Htr3a$-GFP $+$ INs was manually performed with tungsten needles (WPI) in Leibovitz medium (L15; Sigma) under a fluorescence scope (Leica, M165 FC). PSB and CGE tissues were grafted into the PSB region of WT isochronic host slices. Each WT host slice received PSB (right side) and CGE (left side) from $Htr3a$-GFP $+$ donor tissue. After grafting, inserts were directly transferred back to the incubator. Time-lapse imaging of $Htr3a$-GFP $+$ INs derived from the PSB or CGE was performed on cortical slices at two days $in \, vitro$. Inserts were transferred to Fluorodishes (WPI), grafted PSB and CGE $Htr3a$-GFP $+$ were imaged with an inverted confocal microscope (Nikon A1R) equipped for multi-position live imaging (Life Technologies) with a long working distance 20x objective (CFI Plan Fluor ELWD C 20 × / 0.45, Nikon). The microscope incubation chamber temperature was kept at $37\,°C$ with a constant flux ($25 \, l \, h^{-1}$) of 5% CO$_2$ humidified at 96%. 50 μm-thick stacks (3 μm-stepped) were acquired every 10 min during 10–12 h, with resonant laser scanning to reduce toxicity and to avoid bleaching. The first 60 min of movies were removed from analysis to avoid bias in measurements due to settling of the slices in the microscope chamber. For quantification, $Htr3a$-GFP $+$ INs derived from PSB grafts ($n = 47$ cells, $n = 3$ slices from $n = 3$ brains) and CGE grafts ($n = 156$ cells, $n = 3$ slices from $n = 3$ brains) located in the IZ were tracked during a 8 h movie. Migration speed was calculated as the total distance travelled by $Htr3a$-GFP $+$ INs divided by total imaging time. Directional persistence was calculated as the ratio of arithmetic displacement to travelled trajectory length.

**Ectopic $Meis2$ expression in CGE-derived INs.** A plasmid containing the DNA $Meis2$-TurboGFP tag (pCMV6-$Meis2$-tGFP, Origene, clone MG226569) was used as a template to obtain $Meis2$-tGFP tag DNA by gradient PCR. The primers used to amplify the fragment contained 15 base pairs upstream of the binding region, with full homology with the receptor plasmid containing the Dlx5/6 enhancer[59]. The insert and the linearized vector were cloned together using the In Fusion kit method (Clontech).

To label CGE-derived INs, embryos from time-pregnant E14 CD1 mice were electroporated (5 × 50 ms pulse at 45 V) in the VZ of the CGE as previously described[59]. $Dlx5/6$-GFP (1 μg μl$^{-1}$) or $Dlx5/6$-GFP together with $Dlx5/6$-$Meis2$-tGFP (1 and 2 μg μl$^{-1}$, respectively) constructs were delivered using a Picospritzer III microinjection system (Parker). Brains of electroporated pups were processed for analysis at P5. Due to sparse distribution in the cortex, electroporated $Dlx5/6$-GFP $+$ INs were counted in upper (layer 1–4) and lower layers (L5-6) of cortical regions including motor and somatosensory cortices, as well as in the underlying white matter region. In the ectopic $Meis2$ condition, $Dlx5/6$-$Meis2$-tGFP $+$ INs were visualized using immunohistochemistry using a polyclonal rabbit anti-tGFP antibody (1:2,000; ThermoScientific). $Dlx5/6$-GFP $+$ INs displaying a nuclear tGFP signal were used for quantification in the ectopic $Meis2$ condition. Detailed counts of cells and brains for each experimental condition are provided in Supplementary Table 14.

**Statistical analysis.** No statistics were used to determine optimal group sample size; however, sample sizes were similar to those used in previous publications from our group and others. No blinding was performed and no samples were excluded from statistical analysis performed on IHC data. Descriptive statistical analyses, unpaired $t$-tests and estimation of variance were performed using GraphPad Prism software, version 6.0.

**Data availability.** RNA-seq data are available in Supplementary Data 1, in a publicly accessible database (GEO repository, accession number GSE90860) and an interactive website (http://dayerlab.unige.ch/htr3a_sc).

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

## Acknowledgements

We thank the Geneva Bioimaging facility, the Geneva Flow Cytometry facility, the Geneva Genomics facility, Greta Limoni and Christiane Aubry for technical assistance, and Tania Vitalis for providing the *Htr3a*-GFP mice. Work in the Dayer and Holtmaat laboratories was supported by a Swiss National Foundation (SNF) Synapsy grant and a SNF (31003A_155896/1) grant for A.D.

## Author contributions

A.D. conceived the project, A.D., S.F., M.N. C.C. designed the experiments, S.F., M.N., C.C., F.M., U.T., L.G. performed the experiments, S.F., J.P., M.N., C.C., F.M., L.T., A.D. analysed the data and S.F., J.P., M.N., C.C., L.T., A.H., D.J, A.D. wrote the manuscript.

## Additional information

**Competing financial interests:** The authors declare no competing financial interests.

