## [Peer Review File · Nature Communications]

Reviewers' comments:

Reviewer #1 (Remarks to the Author):

In this manuscript, Frazer et al. investigate the transcriptional diversity of cortical interneuron subclasses. Dysfunction of these neurons has been linked to neurodevelopmental disorders and therefore understanding the molecular mechanisms underlying their diversity is fundamental to reveal the etiology of these disorders. Here the authors showed that neurons that express the ionotropic serotonin receptor 5HT3A can be grouped into 3 main types during early postnatal development on the basis of transcriptional profiles. One of these types, which the authors suggest originates at the pallial-subpallial boundary, expresses the transcription factor Meis2 and appears restricted to the cortical white matter. Overall these are interesting findings. However there are several issues that need to be addressed.

Major Issues:

1) Identification of Htr3aR IN types

By using an elegant single cell RNAseq approach, the authors revealed the presence of 3 different types of Htr3aR INs at P2 and P5. One question is to what extent are the genes that are driving the clustering analysis being dynamically regulated with respect to birthdate rather than stably expressed in a way that would define a functionally distinct interneuronal subclass. Could the 3 types identified by transcriptional analyses at each developmental stage (P2 and P5) represent different maturation stages due to different birthdates rather than different functional neuronal types? Do the other Meis2-negative transcriptionally defined subclasses also show restricted anatomical distribution?

Transcriptional analysis at P21 and adult stages should be included in the manuscript to confirm the presence or absence of these same subtypes in the adult animal.

2) Integration of Meis2-interneurons

Whereas Meis2-expressing INs are detected in the cortical white matter and deep cortical layers at P21, it is unclear whether these neurons integrate into cortical circuits. Recordings of synaptic currents as well as morphological reconstructions of Htr3aR WM INs would help to determine whether the cells complete differentiation and integrate into local circuits.

3) Embryonic origin of Meis2-expressing WM neurons

The authors showed that Meis2 is expressed in Htr3aR neurons located in the PSPB -but not within the CGE- at e18 suggesting that cortical Htr3aR WM INs have an embryonic origin distinct from other Htr3aR+ that are CGE-derived. An ultrasound-guided retroviral approach targeting the PSPB followed by analysis of Meis2 expression in WM INs at P5 could provide direct evidence that these neurons originate in this region, and also show the relative contribution of this region to the rostral migratory stream. This strategy has been previously used to successfully map the origin of extracortical subplate cells (Pedraza et al. 2014).

Minor issues:

1) How do the subclasses described in this study relate to the ones found by Zeisel et al 2015 and Tasic et al., 2016?

2) A clearer description of the methodology used to anatomically select the brain regions used for cell counts carried at P5 and P21 should be provided. This information is important to ensure that the rostral migratory stream is completely excluded from analysis.

3) The functional relevance of WM interneurons is not discussed in the manuscript. Why is it important to understand the differentiation of these neurons?

Reviewer #2 (Remarks to the Author):

The paper identifies 3 major classes of Htr3a+ interneurons and their precursors during perinatal development by cell classification based on single cell RNA-seq. It discovers and convincingly demonstrates that one of those 3 classes has a distinct molecular signature including transcription factor Meis2. It shows that this type is enriched in white matter and most likely originates from a separate germinal zone: embryonic pallial-subpallial boundary (PSPB).

This is a solid study that utilizes appropriate methods. The data presentation could be improved, the text should avoid overstatements, and the figure legends should be clearer. I recommend it for publication if the revisions below are implemented.

1. Besides the Meis2+ type, there are two other maturing neuron types (not including the precursors). It would be good to address what the other two major classes may be. Do they end up being Vip+ or Ndnf+? Do they express any markers that are common with the ones described in Zeisel et al or Tasic et al.?
2. Does the Meis2+ type correspond to any type found in Zeisel et al or Tasic et al? Meis2+ type is also present in L6, and cortical dissections from the two other studies may have included some white matter. So, the mature Meis2+ type may be present there? Authors should comment on that.
3. Which genes do the precursors express (most of the cells at E18?). It would be good for this to be discussed.
4. In general, there is some redundancy between figures - too many panels for the data that can be shown more succinctly. I suggest compacting them (see below), and making them bigger. It is also good to keep text size consistent across and within figures. In general, the figures should be self-explanatory, the reader should look into the legend only for additional methodological details.
5. The paper contains several overstatements on the contribution that the paper provides. I think these broad statements are misleading as they are setting up the reader to expect more than the paper provides. Examples:
 - a. Title is too broad. I think it should focus on the discovery of the Meis2+ type, and that should be included in the title.
 - b. Abstract "Here we used single-cell transcriptomics to identify the molecular programs ..." You profiled transcriptomes of cells and identified 3 groups/classes/types and followed up on one without any causality analysis. This is good work, but it should not be stated as 'identifying molecular programs'. 'Identifying molecular programs' involves completing a much bigger task than what the study provides.
 - c. Abstract: "Transcriptional and anatomical parcellation of cortical interneurons are developmentally coupled." In the case of Meis2+ type vs. other Htr3a+ types this is the case, but in the case of many other Htr3a+ types it is likely not the case. So, I think this is again an overstatement that should be modified to be focused on the Meis2+ type vs. other Htr3a+ types. Same comment for lines 84-87, lines 219-220, and lines 252-255.
 - d. Line 194: "...and provide a comprehensive database describing the developmental transcriptional dynamics operating in these three main types of Htr3a-GFP+ interneurons as they integrate into cortical circuits." Too strong of a statement. "Provide transcriptomic profiles" is more appropriate.
6. Line 109: "Revealing the lineage progression of Htr3a-GFP+ INs types across time". "Revealing lineage progression" is an overstatement as the study is taking snapshots in time and cannot connect cells from point A to point B. The data is suggestive, but it does not include lineage tracing. So, a more accurate statement would be: Revealing the gene expression changes within the Htr3a-GFP+ INs types across time.
7. Starting with Line 130: too many numbers with error bars listed in text. This is not wrong, but it is not user-friendly, and it would be much more nicely presented within the figure or in a table. Also, all

SEMs have two decimal points. How was the statistical confidence of those established? I recommend (as taught in statistics classes) using a single significant figure for SEM (only a single figure different from zero), and then round up the measurement to that same position. For example: 85.77 {plus minus} 1.51 will become 86 {plus minus} 2; 2.22 {plus minus} 0.35 will become 2.2 {plus minus} 0.4.

8. I suggest using gene name Htr3a and protein Htr3a instead of 5-HT3AR. The 5-HT3AR name can be listed only once when the gene is introduced. Also, the authors should check what is the convention on protein labeling in mice. I don't think they should use MEIS2 but Meis2, and Meis2 for gene/RNA. Same for Htr3a.

9. Fig 1 legend: Use Fluorescence-activated cell sorting instead of fluorescence-assisted cell sorting (FACS).

10. Fig 1a contains unnecessary repetition of Figs 1b-1d, and it is tiny. I suggest removing 1a and using 1b-1d as the description of the experimental process.

11. Fig 1e) legend "Seurat-based t-Distributed Stochastic Neighbor Embedding (t-SNE) cluster analysis identifies 3 distinct clusters of Htr3a-GFP+ INs across postnatal time-points and a E18 cluster." tSNE is not considered cluster analysis. The representation can be called density clustering by t-SNE. But there is no segregation into clusters, which is standard cluster analysis.

12. 1f legend: replace "combined to" with "combined with"

13. Fig 1e, f, g: Redundant: Why is clustering not performed on all cells at the same time, and a single dendrogram shown? No need to show separate dendrograms for different time points. In addition, tSNE dimensionality reduction can be used to display the data using a different approach, but it is not necessary. It would be best to show single co-clustering dendrogram, where there would be two "legend" color bars on top: one color set would designate types in one bar and the other would be time points. In that case, it would be clear that cells from different time points co-cluster together into same major types, but that the proportions vary at different time points.

14. Fig 1h) Why are two different time points shown separately? Join all cells in a single plot. Label types in the colors as presented, and time points as a separate bar with different colors. Label the rows that correspond to genes pointed out in j).

15. Fig 2a - second bar graph: what do the percentages represent (they don't add up to 100% together, or in individual columns)? Why don't the bars add up to 100% overall (like, in Fig 3a, second bar graph)? Maybe something is wrong with the scale? Figure legend should address each graph.

16. Fig 4a - All or some Meis2/GFP double-positive cells should be indicated by arrowheads in magnified views. Same in 4b.

17. All immunofluorescence panels: It is best that all colored genes/transgene labels are shown on top - it is easier to read them then if they are aligned with the left side of the pictures.

18. Fig 4c - WM label should be shown in the big image, and removed from the small one. I had to check what the big image represented in the legend.

19. Line 225-228. "In adulthood, WM interstitial Htr3a-GFP+ INs are heterogeneous, display different types of electrophysiological profiles, express a variety of neuropeptides including VIP, calretinin and calbindin and possess dendrites and axons confined to the WM or extending to deep cortical layers 36." How does this reconcile with your finding of a pretty distinct and homogeneous interneuron type in WM?

20. Line 248: "genome-wide transcriptomics" Either "genome-wide gene expression" or simply "transcriptomics". Transcriptomics is already genome-wide.

21. Line 305: "Clustering analysis". It should be changed into "clustering" or "cluster analysis".

22. Line 331: "P5 brains were then postfixated in 4% PFA overnight while P21 brains were fixed in 4% PFA for 2 hours." Is this correct? Usually the older tissue needs to be stained overnight?

23. Line 341: "rabbit anti-ER81 (1:5000; gift from Silvia Arber)." Has this been published? If yes, add citation. If not, say unpublished.

24. Lines 361-374: Should be presented as a table.

Reviewer #3 (Remarks to the Author):

This paper uses ss RNAseq and immunofluorescence on developing mouse forebrain to demonstrate that 5HT3-expressing cortical interneurons can be parsed into 3 groups. One of these, defined by the expression of Meis3, is predominantly located in the cortical white matter and appears to have a distinct origin (from other 5HT3+ cortical interneurons) in the pallial-striatal angle.

The paper is beautifully written and presented, and the conclusions are well supported by the data. To this reviewer's knowledge, this is one of the finest uses of the ss-RNAseq approach to identifying lineage relationships to anatomical localization of forebrain neurons to date.

A question is whether the demonstration of the lineage-anatomical relationship, without further (and very time and resource requiring) work on exploring the functional relevance of Meis3+ white matter interneurons, is an adequate advance for this journal.

We would like to thank the reviewers for their very helpful comments and advice. Please find attached the amended paper and a detailed response to the reviewers' comments. Text inserted/modified into the amended paper is highlighted yellow.

Reviewer #1

1) One question is to what extent are the genes that are driving the clustering analysis being dynamically regulated with respect to birthdate rather than stably expressed in a way that would define a functionally distinct interneuronal subclass. Could the 3 types identified by transcriptional analyses at each developmental stage (P2 and P5) represent different maturation stages due to different birthdates rather than different functional neuronal types?

We performed an additional clustering analysis on RNAseq data obtained from all pooled cells isolated at the three different developmental ages. PCA analysis indicated that clustering was primarily based on type-identity and secondarily on their developmental age. In addition, the fact that cells belonging to a given type stably express a distinct set of transcription factors in contrast to other types supports the view that the clusters identified in our work represent distinct types of Htr3a+ interneurons rather than different maturation stages. This data is now available as new Supplementary Fig. 3.

These new data are now described in the results section in the following way:

Clustering was primarily based on type-identity and secondarily on developmental age (Supplementary Fig. 3).

2) Do the other Meis2-negative transcriptionally defined subclasses also show restricted anatomical distribution?

Additional experiments were required to answer this interesting question. As shown in Fig. 1, type 3 Htr3a-GFP+ INs expressed high levels of the transcription factor *Nr2f2* (*CouptfII*). We thus used immunohistochemistry for COUPTFII to determine the spatial distribution of COUPTFII+ / Htr3a-GFP+ INs in the cortex. Interestingly, we found that COUPTFII+ / Htr3a-GFP+ INs were preferentially distributed in layer 1 at both P5 and P21 time-points. These data thus support the idea that type 3 Htr3a-GFP+ INs display anatomical parcellation with enrichment in superficial layer 1. These new data are now presented in new figure 5 and an additional result section has been added to the manuscript to describe this observation.

These new data are now described in the results section in the following way:

Htr3a-GFP+ INs expressing COUPTFII are preferentially located in L1

Type 1 *Meis2*-enriched Htr3a-GFP+ INs were largely confined to the WM. In accordance with this finding, *Meis2* expression was not detected in previously identified interneuron subgroups located in the GM and expressing the *Htr3a* transcript^{34,35} (Supplementary Fig. 9). Given the anatomical segregation of type 1 *Meis2*-enriched INs, we next investigated whether other main types of INs displayed evidence for anatomical parcellation. Single-cell RNA-seq indicated that type 3 Htr3a-GFP+ INs were enriched in the transcription factor *Nr2f2* (Fig. 1i). Using IHC to analyze the spatial distribution of Htr3a-GFP+ INs expressing COUPTFII in the cortex, we found that COUPTFII+ / Htr3a-GFP+ INs were preferentially located in layer 1 (L1) at P5 and P21 (Fig. 5a, b). Htr3a-GFP+ INs in the GM have been

divided in largely mutually exclusive subtypes based on the expression of RELN or VIP^{3,16,18,19,34,35}. Interrogation of our single-cell data indicated that type 3 *Nr2f2*-enriched *Htr3a*-GFP+ expressed higher levels of the *Reln* transcript as compared to other types (Fig. 5c). In accordance with this observation, IHC indicated that RELN was preferentially expressed in COUP-TFII+ / *Htr3a*-GFP+ INs as compared to COUPTFII- / *Htr3a*-GFP+ INs (Fig. 5c). In addition, cross-comparison with previously published single-cell RNAseq datasets^{34,35} indicated that type 3 *Htr3a*-GFP+ INs were preferentially enriched in transcripts found in *Reln* expressing subtypes as compared to *Vip* expressing subtypes (Supplementary Fig. 10). Overall, these results further support a coupling between anatomical and transcriptional parcellation of *Htr3a*-GFP+ interneuron types.

In addition to this new result section, sentences (highlighted in yellow) referring to these new data appear in the introduction and discussion of the manuscript.

3) *Transcriptional analysis at P21 and adult stages should be included in the manuscript to confirm the presence or absence of these same subtypes in the adult animal.*

Two recent papers have revealed the transcriptional diversity of cortical interneurons isolated from the gray matter of the somatosensory cortex (Zeisel et al. 2015) and the visual cortex of adult mice (Tasic et al. 2016). We have performed additional analysis on these datasets to determine whether *Meis2* transcripts could be detected in *Htr3a*-expressing subtypes identified in these datasets. Results indicated that *Meis2* transcripts were not enriched in *Htr3a*-expressing cortical interneurons subtypes identified in the Zeisel et al. and Tasic et al. datasets. We have added this information in new Supplementary Fig 9. The absence of enriched *Meis2* expression in *Htr3a*-GFP+ interneurons isolated from the grey matter is in accordance with our finding that *Meis2* is specifically enriched in *Htr3a*-GFP+ interneurons located in the white matter.

We refer to this new Supplementary figure 9 in the results section in the following way:

Type 1 *Meis2*-enriched *Htr3a*-GFP+ INs were largely confined to the WM. In accordance with this finding, *Meis2* expression was not detected in previously identified interneuron subgroups located in the GM and expressing the *Htr3a* transcript^{34,35} (Supplementary Fig. 9).

In adulthood, white matter *Meis2*-enriched *Htr3a*-GFP+ INs are sparsely distributed in the white matter of adult mice. Using the Fluidigm chip system it was technically not possible to load the chips with a sufficient number of FACS-isolated *Htr3a*-GFP+ from the adult white matter in order to have access to the transcriptome of mature WM *Htr3a*-GFP+ interneurons.

4) *Whereas Meis2-expressing INs are detected in the cortical white matter and deep cortical layers at P21, it is unclear whether these neurons integrate into cortical circuits. Recordings of synaptic currents as well as morphological reconstructions of Htr3aR WM INs would help to determine whether the cells complete differentiation and integrate into local circuits.*

We have performed whole-cell recording of *Htr3a*-GFP+ INs located in the white matter and performed morphological reconstructions as well as MEIS2 immunohistochemistry on patched cells. In new figure 3 and new supplementary figure 6 we show that WM *Htr3a*-

GFP+ IN display spontaneous inhibitory and excitatory currents, express MEIS2 and extend processes in the white matter, overlying cortex and subcortical structures such as the striatum.

These new data are now described in the results section in the following way:

Finally, whole-cell recordings of *Htr3a*-GFP+ INs located in the WM combined with IHC, indicated that these neurons displayed spontaneous inhibitory and excitatory currents (Fig. 3d-g, Supplementary Fig. 6), expressed MEIS2 and extended processes in the WM, deep layers of the cortex and subcortical structures such as the striatum (Fig. 3h, Supplementary Fig. 6). Taken together, our data identify MEIS2-expressing *Htr3a*-GFP+ interneurons that remain confined to the WM and that are functionally integrated in neuronal networks.

5) *The authors showed that Meis2 is expressed in Htr3aR neurons located in the PSB -but not within the CGE- at e18 suggesting that cortical Htr3aR WM INs have an embryonic origin distinct from other Htr3aR+ that are CGE-derived. An ultrasound-guided retroviral approach targeting the PSPB followed by analysis of Meis2 expression in WM INs at P5 could provide direct evidence that these neurons originate in this region, and also show the relative contribution of this region to the rostral migratory stream. This strategy has been previously used to successfully map the origin of extracortical subplate cells (Pedraza et al. 2014).*

Even with ultrasound guidance, it would quite difficult to be confident that a viral infection performed *in utero* at E14 specifically targets the PSB, which is a small structure to access in the embryo. We have thus favored an alternative approach by performing isochronic grafts of *Htr3a*-GFP+ INs on E14 cortical slices using PSB or CGE tissue. Following microdissection of the PSB or CGE under a fluorescent scope, isochronic grafts were performed at E14 on slices to determine whether PSB-derived *Htr3a*-GFP+ INs exit the subpallium and migrate in the prospective white matter of the developing pallium. Using this approach, we could directly monitor and compare the migration of PSB-derived and CGE-derived *Htr3a*-GFP+ in the prospective cortex using confocal time-lapse imaging. We found that *Htr3a*-GFP+ INs derived from PSB grafts efficiently migrated in the prospective white matter of the developing cortex and displayed distinct migratory dynamics as compared to *Htr3a*-GFP+ INs derived from CGE grafts. Two supplementary movies are provided to illustrate this point. In addition, using immunohistochemistry, we found that *Htr3a*-GFP+ INs originating from PSB grafts were MEIS2+ and PROX1-, in contrast to *Htr3a*-GFP+ INs originating from CGE, which were PROX1+ and MEIS2-. Finally, based on previous work in field (De Marco et al. Nature 2011), we used *in utero* electroporation with the Dlx5/6 enhancer construct to target CGE-derived INs. Using this approach, we found that ectopic expression of *Meis2* in CGE-derived INs significantly shifts their cortical location in the white matter and deep cortical layers, thus suggesting that *Meis2* regulates laminar positioning of INs. These additional experiments now appear in new Figure 4, new supplementary figure 6, and in supplementary movies 1 and 2.

These new data are now described in the results section in the following way:

To further investigate the origin of WM INs, isochronic grafts derived from the embryonic PSB or CGE were performed on E14 cortical slices (Fig. 4d). At day *in vitro* 2 (DIV2), PSB-derived *Htr3a*-GFP+ INs migrated into the prospective cortical WM and expressed MEIS2

but not PROX1 in contrast to CGE-derived INs (Fig. 4e). In addition, confocal time-lapse imaging revealed that *Htr3a*-GFP+ INs originating from the PSB displayed distinct migratory dynamics as compared to CGE-derived *Htr3a*-GFP+ INs (Fig. 4f, Supplementary movies 1, 2). Finally, *in utero* electroporation at E14 targeted to the CGE was used to ectopically express MEIS2 in CGE-derived INs. Quantification at P5 indicated that CGE-derived cortical INs, ectopically expressing MEIS2, significantly shifted their laminar distribution towards the WM and lower cortical layers as compared to controls (Supplementary Fig. 8), thus suggesting that MEIS2 regulates anatomical parcellation of cortical interneurons.

6) *How do the subclasses described in this study relate to the ones found by Zeisel et al 2015 and Tasic et al., 2016?*

Our analysis performed on *Htr3a*-GFP+ INs isolated from early postnatal cortex identified two major types (type 2 and type 3) of *Htr3a*-GFP+ INs in the grey matter, whereas type 1 INs were found to be confined to the white matter. We thus aimed to determine how the type 2 or type 3 *Htr3a*-GFP+ INs related to the subtypes identified in the Zeisel et al. and Tasic et al. papers. We found that type 3 *Htr3a*-GFP+ INs shared closer transcriptional proximity with reelin/*Ndnf*+ subclasses identified in the Zeisel and Tasic papers as compared with the *Vip*+ subtypes. These data now appear in new Supplementary fig. 10. In addition, this observation is in accordance with the fact that type 3 *Htr3a*-GFP+ INs express enriched levels of reelin as indicated in new Figure 5.

These new data are now described in the results section in the following way:

In addition, cross-comparison with previously published single-cell RNAseq datasets^{34,35} indicated that type 3 *Htr3a*-GFP+ INs were preferentially enriched in transcripts found in *Reln* expressing subtypes as compared to *Vip* expressing subtypes (Supplementary Fig. 10).

7) *A clearer description of the methodology used to anatomically select the brain regions used for cell counts carried at P5 and P21 should be provided. This information is important to ensure that the rostral migratory stream is completely excluded from analysis.*

We have provided more detailed information in the methods section regarding this point:

At P5 and P21, *Htr3a*-GFP+ cells were counted in the primary motor and somatosensory cortices and in the underlying white matter (WM) in coronal sections corresponding to the levels +0.2 mm to -0.1 mm from Bregma. The quantified region did not contain the rostral migratory stream or the SVZ.

8) *The functional relevance of WM interneurons is not discussed in the manuscript. Why is it important to understand the differentiation of these neurons?*

As indicated in point 4, we now provide electrophysiological data indicating that MEIS2-expressing *Htr3a*-GFP+ INs located in the white matter are functionally integrated and extend processes that reach deep cortical layers and subcortical structures. This is in accordance with a previous study showing that *Htra*-GFP+ interneurons in the cortical white matter display synaptic inputs and extend processes in the cortical and subcortical

structures (Von Engelhardt et al. 2011). The functional role of WM *Htr3a*-GFP+ INs in cortical microcircuits will need to be further determined using optogenetic and connectivity approaches.

Regarding this point, we have modified the discussion in the following way:

In adulthood, WM interstitial *Htr3a*-GFP+ INs have been reported to extend processes into the overlying deep cortical layers and subcortical structures such as the hippocampus and striatum³⁶. In addition, patch-clamp recordings indicated that *Htr3a*-GFP+ INs received excitatory and inhibitory inputs from cortical and subcortical structures and inhibited principal cells of the adjacent cortex³⁶. Future studies using optogenetic-based approaches are thus necessary to probe the role of this WM interneuron type in cortical circuit function.

Reviewer #2

1) Besides the *Meis2*+ type, there are two other maturing neuron types (not including the precursors). It would be good to address what the other two major classes may be. Do they end up being *Vip*+ or *Ndnf*+? Do they express any markers that are common with the ones described in Zeisel et al or Tasic et al.?

Our analysis performed on *Htr3a*-GFP+ INs isolated from early postnatal cortex identified two major types (type 2 and type 3) of *Htr3a*-GFP+ INs in the grey matter, whereas type 1 INs were found to be confined to the white matter. We thus aimed to determine how the type 2 or type 3 *Htr3a*-GFP+ INs related to the subtypes identified in the Zeisel et al. and Tasic et al. papers. We found that type 3 *Htr3a*-GFP+ INs shared closer transcriptional proximity with *reelin*/*Ndnf*+ subclasses identified in the Zeisel and Tasic papers as compared with the *VIP*+ subtypes. These data now appear in new Supplementary fig. 10.

These new data are now described in the results section in the following way:

In addition, cross-comparison with previously published single-cell RNAseq datasets^{34,35} indicated that type 3 *Htr3a*-GFP+ INs were preferentially enriched in transcripts found in *Reln* expressing subtypes as compared to *Vip* expressing subtypes (Supplementary Fig. 10).

2) Does the *Meis2*+ type correspond to any type found in Zeisel et al or Tasic et al? *Meis2*+ type is also present in L6, and cortical dissections from the two other studies may have included some white matter. So, the mature *Meis2*+ type may be present there? Authors should comment on that.

Two recent papers have revealed the transcriptional diversity of cortical interneurons isolated from the gray matter of the somatosensory cortex (Zeisel et al. 2015) and the visual cortex of adult mice (Tasic et al. 2016). We have performed additional analysis on these datasets to determine whether *Meis2* transcripts could be detected in *Htr3a*-expressing subtypes identified in these datasets. Results indicated that *Meis2* transcripts were not enriched in *Htr3a*-expressing cortical interneurons subtypes described in the Zeisel et al. and Tasic et al. datasets. We have added this information in new Supplementary Fig 9. The

absence of enriched *Meis2* expression in *Htr3a*-GFP+ interneurons isolated from the grey matter is in accordance with our finding that *Meis2* is largely enriched in *Htr3a*-GFP+ interneurons located in the white matter.

These new data are now described in the results section in the following way:

Type 1 *Meis2*-enriched *Htr3a*-GFP+ INs were largely confined to the WM. In accordance with this finding, *Meis2* expression was not detected in previously identified interneuron subgroups located in the GM and expressing the *Htr3a* transcript^{34,35} (Supplementary Fig. 9).

3) Which genes do the precursors express (most of the cells at E18?).

We have performed additional analysis using SCDE and now provide in the new supplementary data file additional data providing a dynamics account of gene expression patterns at each developmental time-point including E18 and for each main type of interneuron.

4) In general, there is some redundancy between figures - too many panels for the data that can be shown more succinctly. I suggest compacting them (see below), and making them bigger. It is also good to keep text size consistent across and within figures. In general, the figures should be self-explanatory, the reader should look into the legend only for additional methodological details.

We have followed these suggestions and modified figures as recommended below.

5) Title is too broad. I think it should focus on the discovery of the *Meis2*+ type, and that should be included in the title.

In new figure 5, we provide new data indicating that *Meis2*-negative transcriptionally defined subclasses also show restricted anatomical distribution (see our detailed answer in point 7). Given this new information, we propose to maintain this title. However, we propose to modify the title by adding “5-HT_{3A}R-expressing”, in order to restrict our findings to this class of cortical interneurons.

The new title is: Transcriptomic and anatomic parcellation of 5-HT_{3A}R expressing cortical interneuron subtypes revealed by single-cell RNA sequencing

6) Abstract "Here we used single-cell transcriptomics to identify the molecular programs ..." You profiled transcriptomes of cells and identified 3 groups/classes/types and followed up on one without any causality analysis. This is good work, but it should not be stated as 'identifying molecular programs'. 'Identifying molecular programs' involves completing a much bigger task than what the study provides.

We agree with this comment and have modified in the abstract “Molecular programs” with the more appropriate terminology: “gene expression patterns”.

7) **Abstract: "Transcriptional and anatomical parcellation of cortical interneurons are developmentally coupled. " In the case of *Meis2*+ type vs. other *Htr3a*+ types this is the case, but in the case of many other *Htr3a*+ types it is likely not the case. So, I think this is again an overstatement that should be modified to be focused on the *Meis2*+ type vs. other *Htr3a*+ types. Same comment for lines 84-87 , lines 219-220, and lines 252-255.**

As indicated in our answer to point 5, we aimed to determine whether we could detect evidence for anatomical parcellation of the 2 other main type of gray-matter interneurons (type 2 and type 3). As shown in Fig. 1 type 3 *Htr3a*-GFP+ INs expressed high levels of the transcription factor *Nr2f2* (*Couptf2*). We thus using immunohistochemistry for COUPTFII to determine the spatial distribution of COUPTFII+ / *Htr3a*-GFP+ INs. Interestingly, we found that COUPTFII+ / *Htr3a*-GFP+ INs were preferentially expressed in layer 1 at both P5 and P21 time-points. These data thus indicate that COUPTFII+ / *Htr3a*-GFP+ INs display anatomical parcellation with enrichment in superficial layer 1. These new data are now presented in new figure 5 and an additional result section has been added to the manuscript.

These new data are now described in the results section in the following way:

***Htr3a*-GFP+ INs expressing COUPTFII are preferentially located in L1**

Type 1 *Meis2*-enriched *Htr3a*-GFP+ INs were largely confined to the WM. In accordance with this finding, *Meis2* expression was not detected in previously identified interneuron subgroups located in the GM and expressing the *Htr3a* transcript^{34,35} (Supplementary Fig. 9). Given the anatomical segregation of type 1 *Meis2*-enriched INs, we next investigated whether other main types of INs displayed evidence for anatomical parcellation. Single-cell RNA-seq indicated that type 3 *Htr3a*-GFP+ INs were enriched in the transcription factor *Nr2f2* (Fig. 1i). Using IHC to analyze the spatial distribution of *Htr3a*-GFP+ INs expressing COUPTFII in the cortex, we found that COUPTFII+ / *Htr3a*-GFP+ INs were preferentially located in layer 1 (L1) at P5 and P21 (Fig. 5a, b). *Htr3a*-GFP+ INs in the GM have been divided in largely mutually exclusive subtypes based on the expression of RELN or VIP^{3,16,18,19,34,35}. Interrogation of our single-cell data indicated that type 3 *Nr2f2*-enriched *Htr3a*-GFP+ expressed higher levels of the *Reln* transcript as compared to other types (Fig. 5c). In accordance with this observation, IHC indicated that RELN was preferentially expressed in COUPTFII+ / *Htr3a*-GFP+ INs as compared to COUPTFII- / *Htr3a*-GFP+ INs (Fig. 5c). In addition, cross-comparison with previously published single-cell RNAseq datasets^{34,35} indicated that type 3 *Htr3a*-GFP+ INs were preferentially enriched in transcripts found in *Reln* expressing subtypes as compared to *Vip* expressing subtypes (Supplementary Fig. 10). Overall, these results further support a coupling between anatomical and transcriptional parcellation of *Htr3a*-GFP+ interneuron types.

In addition, reference to these new data appear in the abstract, introduction and discussion and are highlighted in yellow in the revised manuscript text.

8) **Line 194: "...and provide a comprehensive database describing the developmental transcriptional dynamics operating in these three main types of *Htr3a*-GFP+ interneurons as they integrate into cortical circuits." Too strong of a statement. "Provide transcriptomic profiles" is more appropriate.**

We agree with comment and have modified the text appropriately.

9) ***Line 109: "Revealing the lineage progression of Htr3a-GFP+ INs types across time". "Revealing lineage progression" is an overstatement as the study is taking snapshots in time and cannot connect cells from point A to point B. The data is suggestive, but it does not include lineage tracing. So, a more accurate statement would be: Revealing the gene expression changes within the Htr3a-GFP+ INs types across time.***

We agree with this comment and have modified the text accordingly.

10) ***Starting with Line 130: too many numbers with error bars listed in text. This is not wrong, but it is not user-friendly, and it would be much more nicely presented within the figure or in a table. Also, all SEMs have two decimal points. How was the statistical confidence of those established? I recommend (as taught in statistics classes) using a single significant figure for SEM (only a single figure different from zero), and then round up the measurement to that same position. For example: 85.77 {plus minus} 1.51 will become 86 {plus minus} 2; 2.22 {plus minus} 0.35 will become 2.2 {plus minus} 0.4.***

We agree with this comment. We have removed the majority of numbers in text and kept only the most important numbers (highlighted in yellow in revised manuscript text). Numbers reflecting percentages all appear in the figure graphs, which contain the mean and the sem values. Finally, for the numbers in the text, we have followed the suggestion to use a single significant figure for mean and SEM.

11) ***I suggest using gene name Htr3a and protein Htr3a instead of 5-HT3AR. The 5-HT3AR name can be listed only once when the gene is introduced. Also, the authors should check what is the convention on protein labeling in mice. I don't think they should use MEIS2 but Meis2, and Meis2 for gene/RNA. Same for Htr3a.***

We have checked the convention at :

<http://www.informatics.jax.org/mgihome/nomen/gene.shtml>. Protein designations follow the same rules as gene symbols, with the following two distinctions: protein symbols use all uppercase letters and protein symbols are not italicized. We will thus follow these rules and for example use MEIS2 for protein expression.

Following previous publications such as in Chittajallu et al. Nat. Neurosci. 2013 (<http://www.nature.com/neuro/journal/v16/n11/full/nn.3538.html>) we will refer to 5-HT_{3a}R-containing interneurons as a distinct class of interneurons and to *Htr3a*-GFP+ interneurons in experimental procedures using the mouse line.

12) ***Fig 1 legend: Use Fluorescence-activated cell sorting instead of fluorescence-assisted cell sorting (FACS).***

We agree with this comment and have modified the text accordingly.

13). ***Fig 1a contains unnecessary repetition of Figs 1b-1d, and it is tiny. I suggest removing 1a and using 1b-1d as the description of the experimental process.***

We agree with this comment and have modified figure 1 accordingly.

14). Fig 1e) legend "Seurat-based t-Distributed Stochastic Neighbor Embedding (t-SNE) cluster analysis identifies 3 distinct clusters of Htr3a-GFP+ INs across postnatal time-points and a E18 cluster." tSNE is not considered cluster analysis. The representation can be called density clustering by t-SNE. But there is no segregation into clusters, which is standard cluster analysis.

We agree with this comment and have modified the text accordingly.

12) 1f legend: replace "combined to" with "combined with »

We agree with this comment and have modified the text accordingly.

13) Fig 1e, f, g: Redundant: Why is clustering not performed on all cells at the same time, and a single dendrogram shown? No need to show separate dendrograms for different time points. In addition, tSNE dimensionality reduction can be used to display the data using a different approach, but it is not necessary. It would be best to show single co-clustering dendrogram, where there would be two "legend" color bars on top: one color set would designate types in one bar and the other would be time points. In that case, it would be clear that cells from different time points co-cluster together into same major types, but that the proportions vary at different time points.

We performed an additional clustering analysis on RNAseq data from all pooled cells isolated at the three different developmental ages. PCA analysis indicated that clustering was primarily based on type-identity and secondarily on their developmental age. This data is now available as new Supplementary Fig. 3. We propose to maintain the clustering showed in panel f since this clustering allowed us to attribute cells to a given type. In addition, panel f shows that the clustering stability of major types of INs increases with developmental time.

These new data are now described in the results section in the following way:

Clustering was primarily based on type-identity and secondarily on developmental age (Supplementary Fig. 3).

14) Fig 1h) Why are two different time points shown separately? Join all cells in a single plot. Label types in the colors as presented, and time points as a separate bar with different colors. Label the rows that correspond to genes pointed out in j).

The two time-points are shown separately because SCDE analysis revealed type-enriched genes that changed with developmental time progression. The full SCDE gene expression dynamics for each time point and for each main IN type is available in Supplementary data file. We labeled the row for *Meis2* in fig 1h, given that this gene is the focus of the paper. Other genes shown in panel j were not labeled in the fig1h heatmap given that some of them not belong to the top 50 enriched genes displayed in fig1h. The dynamics and violin plots of all enriched genes displayed in fig.1h are displayed in Supplementary fig. 4. The complete list of type-enriched genes can be found in supplementary tables 1-13 and in Supplementary data file.

15) Fig 2a - second bar graph: what do the percentages represent (they don't add up to 100% together, or in individual columns)? Why don't the bars add up to 100% overall (like, in Fig 3a, second bar graph)? Maybe something is wrong with the scale? Figure legend should address each graph.

Thank you very much for spotting this error. The scale has been modified accordingly and cells now add up to 100%.

16) Fig 4a - All or some Meis2/GFP double-positive cells should be indicated by arrowheads in magnified views. Same in 4b.

Arrowheads have been added in figure 4a and 4b.

17). All immunofluorescence panels: It is best that all colored genes/transgene labels are shown on top - it is easier to read them then if they are aligned with the left side of the pictures.

We agree with this comment and have modified figures accordingly.

18) Fig 4c - WM label should be shown in the big image, and removed from the small one. I had to check what the big image represented in the legend.

We agree with this comment and have modified figures accordingly.

19) Line 225-228. "In adulthood, WM interstitial Htr3a-GFP+ INs are heterogeneous, display different types of electrophysiological profiles, express a variety of neuropeptides including VIP, calretinin and calbindin and possess dendrites and axons confined to the WM or extending to deep cortical layers 36." How does this reconcile with your finding of a pretty distinct and homogeneous interneuron type in WM?

Type 2 and type 3 grey-matter Htr3a-GFP+ interneuron types are molecularly diverse as described at later stages in the Zeisel and Tasic papers and in Supplementary figure 4. Similarly, and as shown in Supplementary Fig. 4, we provide evidence for subtype diversity in type 1 WM interneurons.

20) Line 248: "genome-wide transcriptomics" Either "genome-wide gene expression" or simply "transcriptomics". Transcriptomics is already genome-wide.

We agree with this comment and have modified the text accordingly.

21) Line 305: "Clustering analysis". It should be changed into "clustering" or "cluster analysis".

We agree with this comment and have modified the text accordingly.

22). Line 331: "P5 brains were then postfixed in 4% PFA overnight while P21 brains were fixed in 4% PFA for 2 hours." Is this correct? Usually the older tissue needs to be stained overnight ?

This is an error. P21 brains were indeed postfixed in 4% PFA overnight and we have modified the text accordingly.

Brains were then postfixed in 4% PFA O.N.

23). Line 341: "rabbit anti-ER81 (1:5000; gift from Silvia Arber)." Has this been published? If yes, add citation. If not, say unpublished.

The citation for ER81 been added (reference 58).

58. Arber, S., et al. ETS gene Er81 controls the formation of functional connections between group Ia sensory afferents and motor neurons. *Cell* 26, 485-98 (2000).

24). Lines 361-374: Should be presented as a table.

We agree with this comment and have modified this method section by adding a reference to a new Supplementary table 14 which provides the details of cells/brains for each experimental condition.

Reviewer #3

This paper uses single cell RNAseq and immunofluorescence on developing mouse forebrain to demonstrate that 5HT3-expressing cortical interneurons can be parsed into 3 groups. One of these, defined by the expression of Meis3, is predominantly located in the cortical white matter and appears to have a distinct origin (from other 5HT3+ cortical interneurons) in the pallial-striatal angle.

The paper is beautifully written and presented, and the conclusions are well supported by the data. To this reviewer's knowledge, this is one of the finest uses of the ss-RNAseq approach to identifying lineage relationships to anatomical localization of forebrain neurons to date.

We thank reviewer 3 for the positive evaluation of our work.

REVIEWERS' COMMENTS:

Reviewer #1 (Remarks to the Author):

The main message in this manuscript by Frazer et al. remains interesting. Concerns were largely addressed.

Remaining concerns:

- a. The experimental "n" should be added to the figure legends.
- b. In Figure 4d, it is unclear how many brains were analyzed in the transplant experiments. In the methods, the authors refer to "slices". Did all slices come from the same brain?
- c. In supplementary Figure 8, how many brains were electroporated for the laminar analysis?
- d. Justification for the sample sizes used in the added experiments should be included in the manuscript.
- e. The reelin staining in panel Figure 5d is difficult to visualize. Perhaps inverting the image and generating a white background would help to show the cells more clearly.

Reviewer #2 (Remarks to the Author):

The authors have addressed all of my concerns and I recommend that the paper be accepted for publication.

The only thing I spotted is in Lines 248-249: "Here we find that type 1 Meis2-enriched Htr3a-GFP+ INs identified using single-cell RNA-seq are spatially confined to the cortical WM, whereas type 3 Nr2f2-enriched ..."

"spatially confined": too strong. "largely confined to white matter" or "mostly located in white matter" better. Make sure to make it consistent across the paper.

Reviewer #1

- 1) *The experimental “n” should be added to the figure legends.*

This has been added in all figure legends.

- 2) *In Figure 4d, it is unclear how many brains were analyzed in the transplant experiments. In the methods, the authors refer to “slices”. Did all slices come from the same brain?*

Each slice came from a different brain (n = 3). This is now indicated in the figure legend.

- 3) *In supplementary Figure 8, how many brains were electroporated for the laminar analysis?*

In the control condition n = 5 brains and in the MEIS2 condition n = 7. This is now indicated in the figure legend and in the Supplementary table 14.

- 4) *Justification for the sample sizes used in the added experiments should be included in the manuscript.*

We provide in the methods section the following justification: “sample sizes were similar to those used in previous publications from our group and others”. Indeed, in our previous work (Murthy et al. 2014 Nat. Comm.), significant shifts in the laminar distribution of cortical Htr3a-GFP+ INs were observed using similar sample size.

- 5) *The reelin staining in panel Figure 5d is difficult to visualize. Perhaps inverting the image and generating a white background would help to show the cells more clearly.*

We now provide a new image with a white background.

Reviewer #2

- 1) *The only thing I spotted is in Lines 248-249: “Here we find that type 1 Meis2-enriched Htr3a-GFP+ INs identified using single-cell RNA-seq are spatially confined to the cortical WM, whereas type 3 Nr2f2-enriched ...” “spatially confined”: too strong. “largely confined to white matter” or “mostly located in white matter” better. Make sure to make it consistent across the paper.*

We have made this modification across the paper